# Sialic acid mediated mechanical activation of β2 adrenergic receptors by bacterial pili

Zoe Virion[1,7], Stéphane Doly [2,7], Kusumika Saha[2], Mireille Lambert[2], François Guillonneau [3], Camille Bied[2], Rebecca M. Duke[4], Pauline M. Rudd[4], Catherine Robbe-Masselot[5], Xavier Nassif[1,6], Mathieu Coureuil[1]* & Stefano Marullo [2]*

Meningococcus utilizes β-arrestin selective activation of endothelial cell β2 adrenergic receptor (β2AR) to cause meningitis in humans. Molecular mechanisms of receptor activation by the pathogen and of its species selectivity remained elusive. We report that β2AR activation requires two asparagine-branched glycan chains with terminally exposed N-acetyl-neuraminic acid (sialic acid, Neu5Ac) residues located at a specific distance in its N-terminus, while being independent of surrounding amino-acid residues. Meningococcus triggers receptor signaling by exerting direct and hemodynamic-promoted traction forces on β2AR glycans. Similar activation is recapitulated with beads coated with Neu5Ac-binding lectins, submitted to mechanical stimulation. This previously unknown glycan-dependent mode of allosteric mechanical activation of a G protein-coupled receptor contributes to meningococcal species selectivity, since Neu5Ac is only abundant in humans due to the loss of CMAH, the enzyme converting Neu5Ac into N-glycolyl-neuraminic acid in other mammals. It represents an additional mechanism of evolutionary adaptation of a pathogen to its host.

[1] Inserm, U1151, CNRS UMR 8253, Institut-Necker-Enfants-Malades, Université de Paris, Paris, France. [2] Inserm, U1016, CNRS UMR8104, Institut Cochin, Université de Paris, Paris, France. [3] Plateforme de Protéomique, Université de Paris, Paris, France. [4] NIBRT GlycoScience Group, NIBRT – The National Institute for Bioprocessing Research and Training, Blackrock, Co., Mount Merrion, Fosters Avenue, Dublin, Ireland. [5] CNRS, UMR 8576, Unité de Glycobiologie Structurale et Fonctionnelle (UGSF), Université Lille, 59000 Lille, France. [6] Assistance Publique – Hôpitaux de Paris, Hôpital Necker Enfants Malades, Paris, France. [7] These authors contributed equally: Zoe Virion, Stéphane Doly. *email: mathieu.coureuil@inserm.fr; stefano.marullo@inserm.fr

Meningococcus (*Neisseria meningitidis*) is a human-restricted Gram-negative bacterium resident in the nasopharyngeal mucosa. If meningococci gain access to the bloodstream, they rapidly disseminate, cross the blood–brain barrier (BBB), and invade meninges, causing cerebrospinal meningitis[1]. In case of high bacterial load, infection can rapidly progress leading to peripheral vascular involvement and septic shock, the most severe life-threatening form being *purpura fulminans*[2].

Interaction of *N. meningitidis* with endothelial cells is critical for the initiation of peripheral vascular lesions[3] and for the opening of BBB[4]. This interaction involves long bacterial filamentous structures on the pathogen known as Type IV pili (Tfp), which mediate the attachment of virulent capsulated meningococci to endothelial cells in vitro and in vivo[4,5]. Tfp are made of the assembly in helical fibers of a core pilin subunit, PilE, and of other less abundant (minor) pilins, such as PilV, PilX, or ComP, which are structurally similar to PilE[6,7]. They trigger signaling cascades in host cells leading to the stabilization of bacterial colonies at the endothelial cell surface and the subsequent translocation of bacteria through endothelial barriers[8–10]. Tfp successively interact via PilE and PilV with two host receptors that form a constitutive oligomeric complex in endothelial cells:[11] CD147, which functions as the initial adhesion receptor[12], and the signaling $\beta_2$-adrenergic receptor ($\beta_2$AR[9]), a member of the G protein-coupled receptor (GPCR) family.

A major unresolved question in this context is the molecular mechanism by which a GPCR can be activated by bacterial pilins to transduce a signaling cascade that is normally promoted by cognate receptor ligands. Catecholamines and adrenergic agonists bind to and fully activate $\beta_2$AR by interacting with its orthosteric ligand-binding pocket[13]. This interaction causes receptor coupling to cognate $G_s$ protein, which activates adenylyl cyclases, and also the GPCR kinase (GRK)-dependent recruitment of $\beta$-arrestins, which scaffold signaling cascades and regulate receptor response[14]. In contrast, meningococcal pili do not promote cAMP production in host cells[9] and only induce a GRK-dependent activation of $\beta$-arrestins, which in turn activate a Src-cortactin pathway essential for the stabilization of bacterial colonies under blood flow, and recruit p120-catenin and VE-cadherin depleting them from endothelial junctions, causing the opening of the BBB[9]. The activation by pili is allosteric, as it cannot be inhibited by an orthosteric blocker such as propranolol, and was reported to somehow involve the N-terminal extracellular region of the $\beta_2$AR. Indeed, substituting the N-terminal region of the infection-incompetent angiotensin II receptor AT1R with that of the human $\beta_2$AR produced a chimeric receptor that could be activated by meningococci in vitro[9], suggesting that some direct or indirect *N. meningitidis* interaction with the $\beta_2$AR N-terminus might mediate $\beta$-arrestin-selective signaling.

Growing *N. meningitidis* colonies are submitted to forces exerted by blood flow. Consequently, to cope with hemodynamic forces that oppose attaching to endothelial cells at the initial stages of infection, bacterial adhesion only occurs at low levels of shear stress, which are mostly found in capillaries[15]. Moreover, Tfp-induced signaling triggers host cell plasma membrane reorganization to form filopodia-like structures that come in close contact with bacteria, expanding the interaction surface between the colonies and the endothelium and contributing to their resistance to shear stress[10,11]. In addition to their passive "hooking" role, Tfp are actively involved in the generation of mechanical forces. Early studies in *Neisseria* species demonstrated that pilus retraction powered by the PilT ATPase generates nanonewton forces in vivo allowing bacteria to crawl over surfaces[16,17]. These forces, when applied to the cell surface via pili-coated beads mobilized by optical tweezers, were sufficient to

induce ezrin recruitment under beads[18]. Also, intermittent Tfp-dependent traction forces inside bacterial aggregates were recently reported to play a role in the adaptation of *N. meningitidis* aggregates to the geometry of invaded capillaries[19]. Although Tfp need to bind to host endothelial cell to generate pulling forces, the plasma membrane tether involved in this process has not been identified yet. Previous studies in flow chambers, which recapitulate the hemodynamic conditions found in capillaries, have demonstrated that the $\beta_2$AR is instrumental in the signaling process, leading to stabilizing small meningococcal colonies under flow[9]. A potential mechanistic explanation of these observations would be that the $\beta_2$AR is at the same time a mechano-sensor and a transducing receptor for meningococcal Tfp.

A further central open question related to meningococcus-induced signaling is why this pathogen only infects humans and cannot activate signals in non-human endothelial cells. Several adaptive features have been proposed to participate in the species selectivity of meningococcus, particularly when the initial bacterial load is low in vivo. *Neisseria meningitidis* has established specific interactions with human transferrin, which carries the iron indispensable for bacterial growth[20]. Similarly, the interaction of meningococcus with human complement factor H protects the pathogen from the lytic action of the complement components[21]. However, in reconstituted cell systems in which these proteins are absent, meningococcus signaling only requires the presence of the $\beta_2$AR at the surface of host cells of human origin, provided that at least a surrogate adhesion system permits the initial adhesion of the pathogen[9]. These observations suggest that the restriction of meningococcal signaling for human cells is somehow linked to the mechanism of activation of the $\beta_2$AR.

Here we show that the $\beta_2$AR functions as a mechano-sensor for Tfp and that its allosteric activation by meningococcus is caused by mechanical traction forces. These forces are applied via two asparagine-branched glycan chains with terminally exposed *N*-acetyl-neuraminic acid (sialic acid (NANA), Neu5Ac) residues in the receptor extracellular extremity, independently of surrounding amino-acid residues. Because of the lack of the enzyme converting Neu5Ac into *N*-glycolyl-neuraminic acid (Neu5Gc), human NANA differs from that of other mammals. The requirement of Neu5Ac for $\beta_2$AR mechanical activation thus appears to be essential in the pathogen selectivity for human cells and represent an additional example of evolutionary adaptation of a pathogen to its hosts.

## Results

**Tfp activate $\beta_2$AR via interaction with receptor N-glycans.** Meningococcus can activate in vitro an infection-incompetent AT1R in which the N-terminus is substituted by that of the human $\beta_2$AR, suggesting that this region of the $\beta_2$AR is essential for triggering receptor signaling[9]. To further demonstrate that meningococcal pilins PilV and PilE specifically interact with the $\beta_2$AR N-terminus, we used a homogeneous time-resolved fluorescence energy transfer (HTRF)-based assay in intact cells (Fig. 1a). Human HEK-293 (human embryonic kidney-293) or baby hamster kidney (BHK) cells were transfected with a construct coding for the HA epitope, followed by the first 27 amino-acid residues of the human $\beta_2$AR, corresponding to the N-terminal extracellular region (EC1), the first transmembrane domain (TM1), and the first residues of the first intracellular loop, fused in phase upstream of green fluorescent protein (GFP) (identified as HA-EC1-TM1$\beta_2$AR-GFP). The signal of GFP fluorescence was used to control that the expression level of this construct was equivalent in the various assays (Supplementary Fig. 1). Cells were first incubated with excess recombinant pilins containing a 6×His N-terminal tag: PilV, PilE, or ComP, a minor

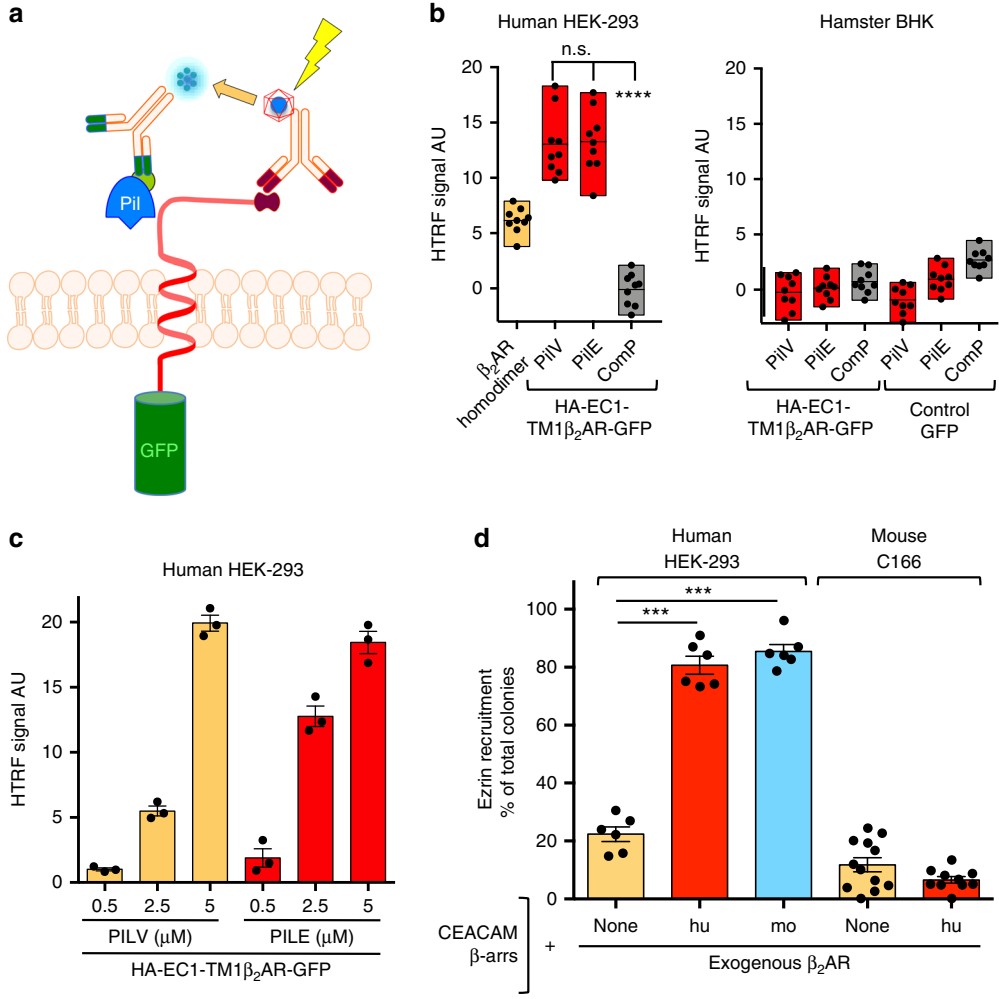

**Fig. 1** $\beta_2$AR activation involves the direct interaction of pilins with receptor N-terminus. **a** Diagram of the HTRF assay. Cells expressing comparable amounts (GFP signal) of the HA-EC1-TM1$\beta_2$AR-GFP construct were incubated with excess recombinant pilins (Pil) containing a 6×His N-terminal tag. Anti-HA-Tag antibodies coupled to the cryptate donor and anti 6×His-Tag antibodies coupled to the acceptor fluorophore were then added for 1 h before excitation at 320 nm and reading (FRET emission at 665 nm) in a Mithras2 Lysogeny Broth (LB) 943 HTRF reader. **b** HTRF signal obtained with PilV, PilE, and ComP (a non-signaling control pilin). The positive control ($\beta_2$AR homodimer) was obtained in HEK-293 cells expressing wild-type HA-tagged $\beta$2-adrenoceptor incubated with a mixture of HTRF donor and acceptor antibodies both directed against the HA-tag, since previous studies reported that $\beta_2$ARs expressed in HEK-293 cells form constitutive homodimers[59]. Some experiments were conducted in hamster BHK cells expressing the same amount of surface HA-EC1-TM1 $\beta_2$AR-GFP or control GFP. **c** The HTRF signal increased with the amount of incubated binding pilin. **d** *Neisseria meningitidis*-promoted signaling assays were conducted in the indicated human or mouse cells in vitro as reported[9]. The readout of the assay is the translocation under bacterial colonies of ezrin, a downstream event of the signaling cascade promoted by the pathogen[60]. Forty eight hours after transfection, HEK-293 cells were infected with bacteria, washed, fixed, and stained with Alexa-conjugated anti-ezrin antibodies. DAPI was used to stain nuclei and bacterial DNA before analysis with a fluorescence microscope. In this experiment, the adhesion of piliated non-capsulated Opa+ (2C43 SiaD- Opa+) meningococci onto HEK-293 cells was achieved by the expression of exogenous human CEACAM-1, a natural receptor for Opa molecules. Signaling in HEK-293 cells, leading to ezrin translocation beneath bacterial colonies, was obtained with exogenous human (hu) or mouse (mo) $\beta_2$AR (expressed at the same concentration, controlled by the YFP signal, except for the control, "none") and $\beta$-arrestins 1 and 2 ($\beta$-arrs). The absence of meningococcus signaling observed in C166 cells was also found with mouse 3T3 fibroblasts (not shown). Data corresponding to at least two independent experiments in triplicate were analyzed by one-way ANOVA; ****$p < 0.0001$, ***$p < 0;001$, n.s.: nonsignificant

pilin not involved in meningococcal signaling and used as a negative control[9,12]. Anti-HA-Tag antibodies coupled to the HTRF donor and anti 6×His-Tag antibodies coupled to the HTRF acceptor were then added for 1 h before excitation and reading. In this assay a FRET signal can only be recorded if the donor and the acceptor are located at a distance of 10 nm or below. A dose-dependent specific HTRF signal was measured with recombinant His-PilV and His-PilE in human cells expressing controlled levels of HA-EC1-TM1$\beta_2$AR-GFP (Fig. 1b, c), whereas no signal was observed with His-ComP. These data indicate a direct binding of PilE and PilV to the extracellular N-terminal region of the human

$\beta_2$AR. Since meningococcus only signals in human cells[1], we compared human and mouse $\beta_2$AR in a reconstituted assay in vitro[9], in the attempt of delineating the possible role of specific amino-acid residues of $\beta_2$AR N-terminus (Supplementary Fig. 2) in receptor activation by bacterial pili. While human HEK-293 cells reconstituted with mouse $\beta_2$AR could be activated by *N. meningitidis* in vitro, the induced signaling being illustrated by ezrin recruitment under colonies as endpoint[9], mouse endothelial cells expressing exogenous human $\beta_2$AR were not activated by the pathogen even in the case of human CEACAM-1-promoted mock adhesion (Fig. 1d). Consistently, no specific binding signal could

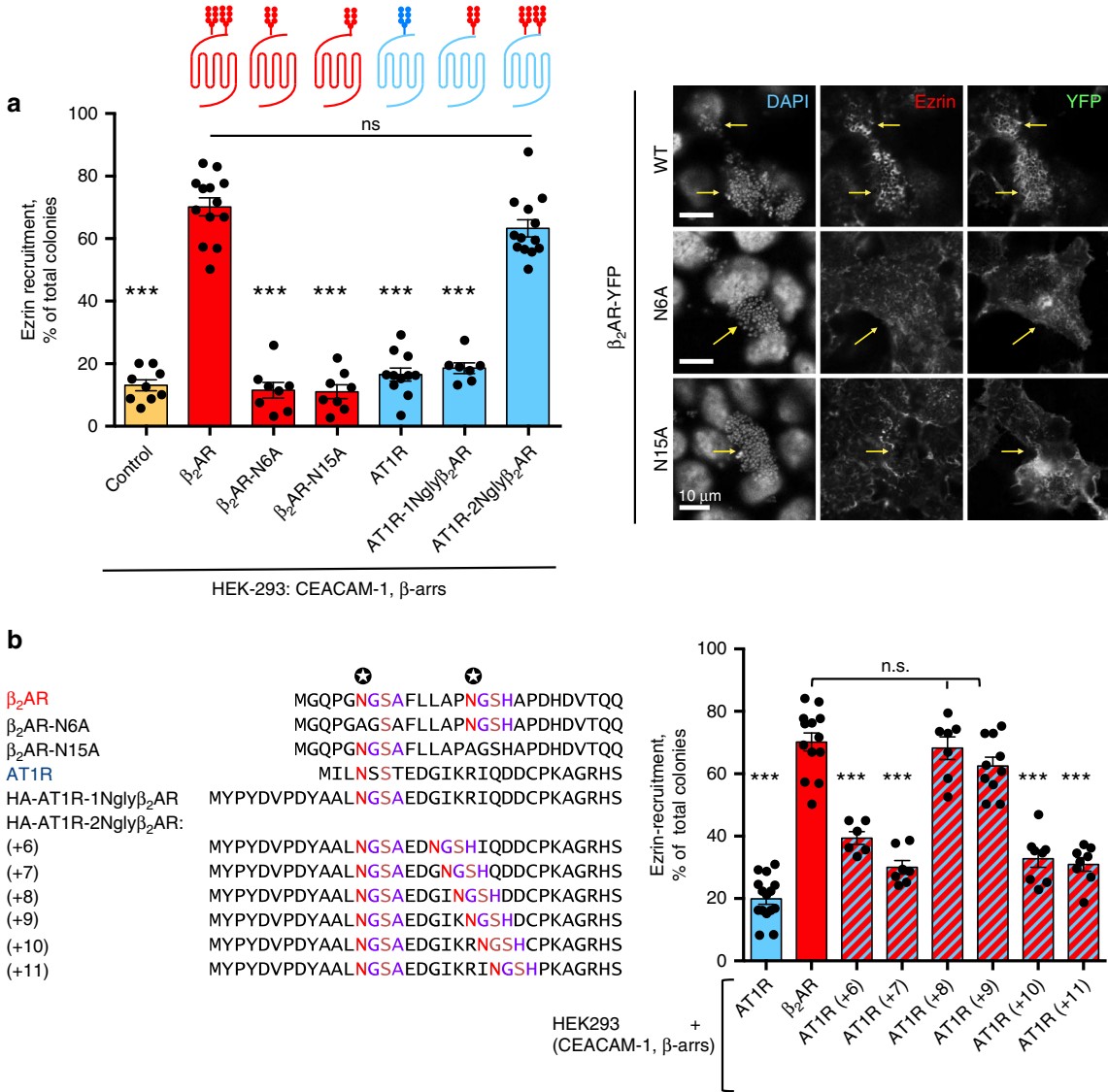

**Fig. 2** Role of glycan chains in meningococcus-promoted β2AR signaling. **a**, Left panel: ezrin recruitment assays were as in Fig. 1d; the percentage of ezrin-recruiting colonies was quantified in controls and in cells expressing human CEACAM-1, β-arrestins (β-arrs) and YFP-tagged forms of: wild-type β2AR, mutant β2AR lacking either N-glycosylation site (β2AR-N6A and β2AR-N15A), wild-type angiotensin AT1R, and mutant AT1R expressing either the first β2AR N-glycosylation signal (AT1R-1Nglyβ2AR), or both signals (AT1R-2Nglyβ2AR (+9)). Red drawings represent the β2AR polypeptide and its N-glycan chains. The blue diagram represents the AT1R polypeptide with its single N-glycan chain. In mutant AT1R receptors, the N-glycan chains branched on the N-glycosylation signals from the β2AR are represented in red. Histograms represent mean values ± SEM ($n = 6$) (***$p < 0.001$ control or mutants vs. wild-type β2AR, ANOVA with Bonferroni multiple comparison test; n.s.: not significant difference). Right panel: representative IF images of the experiment; arrows indicate the position of bacterial colonies; ezrin indicates the labeling with anti-ezrin antibodies and YFP corresponds to the signal of YFP-tagged receptors. **b** HEK-293 cells expressing CEACAM, β-arrestins, and the indicated receptors were prepared, infected, and processed as in **a**. Left panel: sequence alignment of receptor N-termini (constructs used in **a**, **b**). Asparagine residues used for glycan branching on wild-type β2AR are marked by stars. HA-AT1R-1Nglyβ2AR and HA-AT1R-2Nglyβ2AR correspond to AT1R mutants expressing one or both β2AR N-glycosylation signals, respectively; the numbers within parentheses correspond to the distance between N-glycosylation signals, (+9) being the distance found in wild-type β2AR N-terminus; β2AR-N6A and β2AR-N15A are mutant β2ARs lacking either N-glycosylation site. Right panel: role of the distance between N-glycosylation sites on meningococcus-promoted ezrin recruitment. Histograms represent mean values ± SEM; ANOVA with Bonferroni's multiple comparison test: ***$P < 0.001$, n.s.: not significant, for the comparison with wild-type β2AR

be measured with the HTRF assay if the human β2AR N-terminus was expressed in hamster cells (Fig. 1b). Thus, the capacity of meningococci to bind to and activate the β2AR in human cells is independent of the amino-acid sequence and involves other factors. Since the β2AR N-terminus is glycosylated due to the presence of two asparagine-dependent glycosylation sites[22], we examined whether these N-glycan chains might be important for receptor activation by *N. meningitidis*. Alanine substitution of

asparagine residues at position 6 or 15 of the β2AR, to suppress either glycan chain, dramatically reduced ezrin recruitment under meningococcal colonies to basal levels (Fig. 2a), although both receptor mutants were correctly targeted to the cell surface and responsive to agonist activation (Supplementary Fig. 3). The AT1R, which is not activated by meningococcus, only contains one glycosylation site in its terminus. Thus, in a complementary gain-of-function approach, two glycosylation consensus sequences

with asparagine residues separated by nine residues as in the $\beta_2$AR were created by mutagenesis of the angiotensin AT1R complementary DNA (cDNA). Co-expression of this HA-AT1R-2Ngly$\beta_2$AR (+9) mutant with $\beta$-arrestins reconstituted *N. meningitidis*-promoted signaling in vitro, contrasting with the lack of effect of the replacement of the AT1R N-glycosylation consensus sequence with the first of the $\beta_2$AR (HA-AT1R-1Ngly-$\beta_2$AR) (Fig. 2a). These data suggest that both glycan chains of the $\beta_2$AR are necessary and sufficient to support receptor activation by the pathogen, whereas the activation of a plasma membrane receptor via the engagement of its glycan chain(s) usually also involve the protein backbone[23]. We next investigated the importance of the distance between glycan chains in receptor activation (Fig. 2b). While maintaining the first N-glycosylated site at a constant position, the second site (NGSH) was introduced in the AT1R N-terminus at a distance of 6 to 11 residues from the first site. In addition to HA-AT1R-2Ngly$\beta_2$AR (+9), only the AT1R mutant where the glycosylation sites were separated by eight residues (HA-AT1R-2Ngly$\beta_2$AR (+8)) promoted effective meningococcal activation in host cells. Altogether, these data demonstrate that Tfp-promoted signaling requires two glycan chains, at a definite distance in the N-terminus of the $\beta_2$AR, independently of surrounding amino-acid residues.

**NANA residues are essential for $\beta_2$AR activation**. The above data indicate that a particular distance of $\beta_2$AR glycan chains is required for full interaction with meningococcal Tfp and subsequent receptor activation. Lectins were next used as inhibitors of receptor activation in an in vitro assay to identify the sugars that are specifically involved in this process (Fig. 3). This assay, performed with the same procedure as in reconstituted HEK-293 cells, was conducted instead on human hCMEC/D3 cells, which retain main features of primary brain micro-vessel endothelial cells, express endogenous CD147, $\beta_2$AR, and $\beta$-arrestins, and represent a validated model of meningococcal infection[8]. Confirming the prominent role of $\beta_2$AR in mediating meningococcus-promoted signaling, the deletion of the $\beta_2$AR gene in this cell line markedly inhibited ezrin recruitment under bacterial colonies (Supplementary Fig. 4). Among 25 tested lectins, 4 displayed highly significant inhibition; 2 of them, *Maackia amaurensis-I* (Mal-I) and wheat germ agglutinin (WGA), almost completely blocked meningococcus-induced signaling (Fig. 3a, b) in a dose-dependent manner (Fig. 3c). Control experiments were conducted to rule out the hypothesis that the lectin inhibitory effect could depend on the induction of some $\beta_2$AR endocytosis (Supplementary Fig. 5). Mal-I was reported to bind to Neu5Ac $\alpha$(2,3) galactose $\beta$(1,4)GlcNAc[24], whereas WGA binds to both the GlcNAc-GlcNAc (chitobiose) core and terminal NANA[25]. Succinylated WGA, which does not bind to glycans containing sialic acid[25], did not inhibit meningococcus signaling (Fig. 3a, b). Moreover, in vitro meningococcal colonization of hCMEC/D3 cells under shear stress, which can be repressed by depleting the $\beta_2$AR[9], was also inhibited by Mal-I (Supplementary Fig. 6). This assay recapitulates one of the pathophysiological properties of meningococcus, namely the induction of a signaling pathway, which lead to the stabilization of growing colonies under shear stress. Together, the above findings indicate that NANAs participate in $\beta_2$AR activation by the pathogen. Consistently, selective inhibition of sialyl transferases with 3Fax-Peracetyl-Neu5Ac (3FP-Neu5Ac) was sufficient to abolish meningococcus-induced recruitment of $\beta$-arrestin-GFP (Fig. 3d, e) and downstream signaling (i.e., ezrin recruitment, Fig. 3d, f, g) in human endothelial cells. The observation that both glycan chains of the $\beta_2$AR are necessary to support receptor activation and that spacing of the glycans in this context is critical (see Fig. 2) suggests that both

glycan residues in question are sialylated. To address this issue, we took advantage of a trypsin cleavage site at the end of the $\beta_2$AR N-terminus to measure Neu5Ac present in isolated glycosylated receptor N-terminal fragments (Fig. 4, procedure outlined in Supplementary Fig. 7). The N-termini of HA-tagged wild-type (wt)-$\beta_2$AR (+11)-$\beta_2$AR, N6A-$\beta_2$AR, and N15A-$\beta_2$AR released from tryptic digestions of transfected HEK-293 cells were immunoprecipitated with anti-HA antibody-coated beads (Fig. 4a, b). The identity of the fragments recognized by the anti-HA antibodies was confirmed by mass spectrometry (MS) (Fig. 4c). NANAs present in the glycan chains were then released by heating the samples in acidic conditions and quantified by a colorimetric assay (Fig. 4d). The amount of NANA contained in the N-termini of N6A-$\beta_2$AR and N15A-$\beta_2$AR was roughly half of that present in wt-$\beta_2$AR or (+11)-$\beta_2$AR. These data strongly support the conclusion that both glycan chains at positions 6 and 15 can individually undergo terminal sialylation and that both glycan chains from wt-$\beta_2$AR and (+11)-$\beta_2$AR are similarly sialylated.

**NANAs participate in pathogen species selectivity**. Neu5Ac and Neu5Gc, which differ by a single oxygen atom[26], are the predominant NANAs in mammals[27]. Neu5Gc is synthesized from Neu5Ac by the cytidine monophosphate-*N*-acetylneuraminic acid hydroxylase (CMAH)[28]. Only Neu5Ac is present in humans, because of CMAH genetic inactivity[26], while Neu5Gc is predominant in other mammals. To investigate whether the exposed Neu5Ac in the N-glycan chains might contribute to species specificity of meningococcal interaction with the $\beta_2$AR, CMAH was deleted in C166 mouse endothelial cells (Supplementary Fig. 8) to produce a mouse cell clone that only expresses human-type Neu5Ac-containing glycans. To demonstrate the enrichment in Neu5Ac in the *CMAH*-knockout (KO) clone, we took advantage of WGA selectivity for Neu5Ac compared to Neu5Gc[29]. Precipitation assays were performed with WGA-coated agarose beads from extracts of wt and *CMAH*-KO cells expressing exogenous $\beta_2$AR-GFP. The glycosylated form of $\beta_2$AR-GFP was substantially more abundant in precipitations from C166-*CMAH*-KO cells providing indirect evidence that these cells contain a higher proportion of the Neu5Ac form than parental C166 (Supplementary Fig. 8e). Experiments were then performed using a piliated non-capsulated Opa + *N. meningitidis* strain, and CECAM-transfected mouse endothelial cells to allow meningococcal adhesion (see Fig. 2a). In parental C166 cells expressing exogenous human YFP-tagged $\beta_2$AR, the receptor was not redistributed under bacterial colonies, contrasting with the significant accumulation that could be quantified in *CMAH*-KO cells (Fig. 5a). Consistently, full or partial ezrin recruitment was observed in C166 clones lacking CMAH, contrasting with the absence of recruitment in the parental control C166 cell line (Fig. 5b). Furthermore, pre-incubation with the selective agonist isoproterenol, to induce endocytosis of the endogenous $\beta_2$AR, inhibited this signaling, as observed previously in human endothelial cells. These data indicate that the absence of CMAH indeed contributes to the species selectivity of meningococcus for humans.

**Pulling forces applied on glycan chains activate the $\beta_2$AR**. We next aimed at determining how pili binding can induce allosteric $\beta$-arrestin-selective activation of the $\beta_2$AR. Bacteria growing at the cell surface of endothelial cells are permanently submitted to forces exerted by blood flow. In addition, pilus retraction, powered by the PilT ATPase, generates nanonewton forces[16,17] independently of the hemodynamic flow. We hypothesized that mechanical forces applied via Tfp to receptors might produce

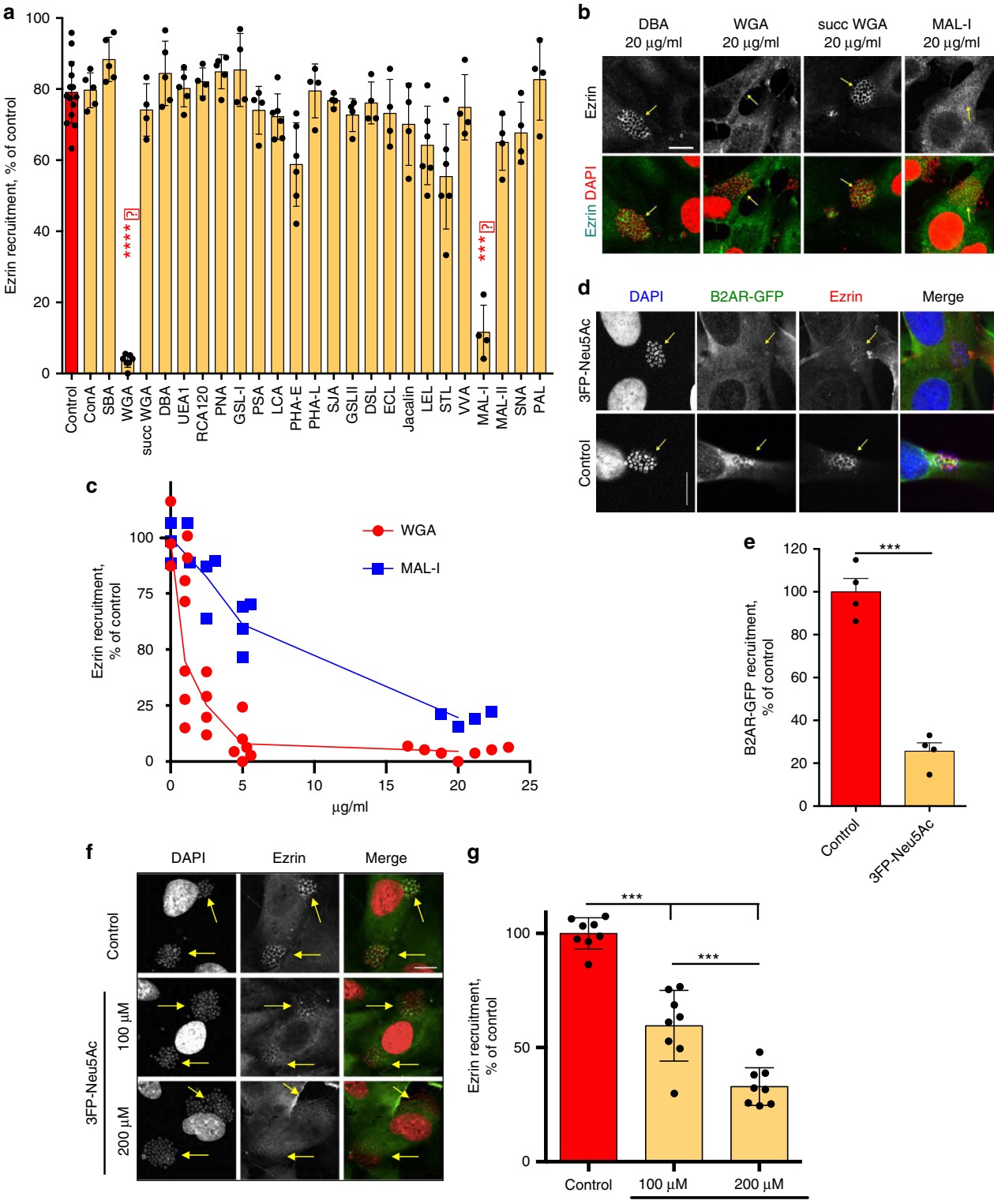

**Fig. 3** Lectins and sialyl-transferases inhibitors prevent meningococcus signaling. hCMEC/D3 cells were incubated for 1 h with the indicated lectins (full names in Supplementary Table 1) at a concentration of 20 μg/ml. Cells were then washed, infected with wild-type *N. meningitidis*, and processed for the analysis of ezrin recruitment under bacterial colonies. **a** Histograms correspond to mean values ± SEM. Data were analyzed by ANOVA with Dunnett's multiple comparisons test. ***$P < 0.001$, ****$p < 0.0001$, control (no lectin) vs. incubation with the indicated lectin. **b** Representative immunofluorescence images. HCMEC/D3 cell nuclei and bacterial DNA (small, grouped dots) are labeled with DAPI; anti-ezrin antibodies stained ezrin. Bar: 10 μm. **c** Dose-dependent inhibition of signaling by WGA and Mal-I. **d, e** Inhibition of β-arr2-GFP recruitment under bacterial colonies by the selective inhibition of sialyl-transferases with 3Fax-Peracetyl Neu5Ac (3FP-Neu5Ac). **d** Representative pictures of the dose–response experiment in **e**, bar: 10 μm, ***$p < 0.001$, $n = 3$. **f, g** Inhibition of ezrin recruitment under colonies by 3FP-Neu5Ac. **f** Representative pictures of the dose–response experiment in **g**, bar: 10 μm, ***$p < 0.001$, $n = 3$ (ANOVA)

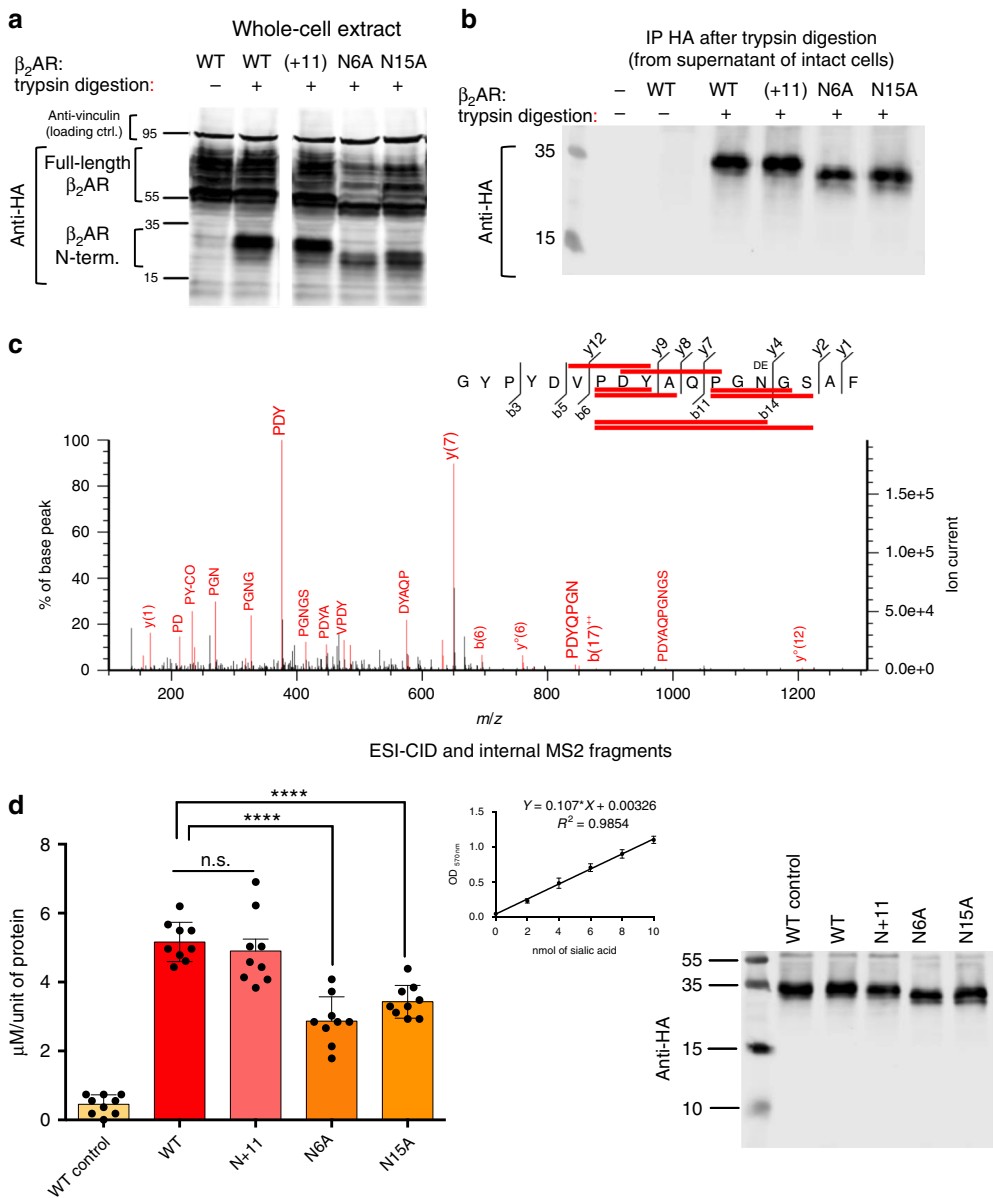

**Fig. 4** Sialylation of N-glycan chains at positions 6 and 15 of the β₂AR N-terminus. **a** Western blot analysis of whole-cell lysates from HEK-293 cells expressing wt or mutant HA-tagged β₂AR (description in Supplementary Fig. 7a). Intact cells were submitted to trypsin digestion (see Methods) to cleave the N-terminus of surface receptors. After the addition of Laemmli buffer directly to the mixture and PAGE separation, WB membranes were probed with the indicated antibodies. The 36-residue glycosylated N-termini produced by the tryptic digestion migrate faster than the corresponding un-cleaved receptors. The N-termini from N6A and N15A β₂AR, which contain only one Asn-branched glycan chain display a lower apparent MW than wt or (+11) β₂AR. **b** Supernatants of the same cells submitted to trypsin digestion were incubated with anti-HA-tag-coated beads to immunoprecipitate the released N-termini. **c** Immunoprecipitated N-termini were additionally digested with PNGase and chymotrypsin to generate glycan-free peptides for MS analysis. The shown chromatogram corresponds to the analysis of the wt β₂AR N-terminus. Spectrum annotation (b- and y-ion series) was done according to nomenclature of peptide fragmentation[61]. Indicated matched internal fragments (red bars) correspond to higher intensity peaks, mostly proline-starting fragments[62] produced by higher energy collisional dissociation (HCD)-type high-resolution mass spectrometer. Indeed, the presence of proline residues causes increased peptide fragility at their N-terminal side during collisionnal-induced decay (CID)[63]. The ᴰᴱN residue corresponds to the change of an asparagine residue into an aspartic acid due to the enzymatic (PNGase) removal of N-linked glycans. **d** Receptor N-termini bound to HA-coated beads were heated in acidic conditions to release Neu5Ac from the glycan chains; liberated sialic acid was quantified by a colorimetric assay (see Methods). Left panel: amount of sialic acid released from the indicated N-termini. "WT control" corresponds to a sample of wt β₂AR bound to HA breads maintained in PBS buffer without acidic treatment. Values of Neu5Ac obtained with the assay were calibrated with the actual protein concentration of the N-terminal fragments determined by quantifying WB images (LI-COR software) (ANOVA: ****$p < 0.0001$, 9 samples for each condition, prepared from 3 independent batches of transfected cells). Middle panel: standard curve. Right panel: representative WB used to quantify the indicated N-terminal fragments. n.s., not significant

their activation. To address this hypothesis, in a dedicated experimental protocol designed to minimize the exogenous forces applied on bacteria (see Methods), wt and mutant meningococci deficient in PilT activity (ΔT[30]) were compared for their capacity to induce signaling in hCMEC/D3 cells. Supporting the contribution of pilus retraction in receptor activation, ezrin recruitment was significantly impaired under ΔT meningococci compared to wt under basal, static conditions (Fig. 6a, b). In

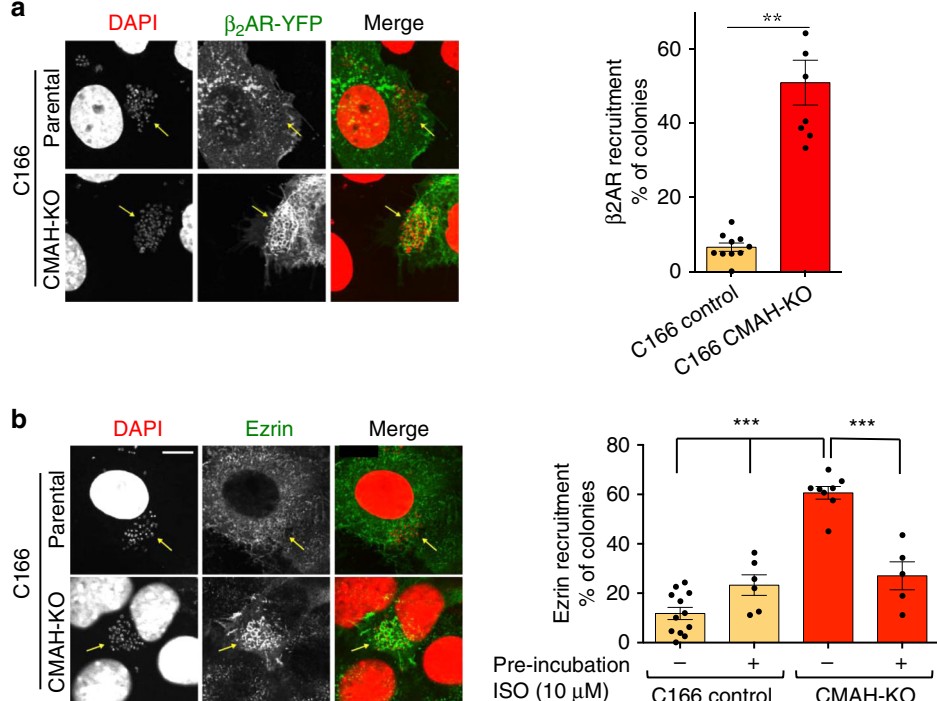

**Fig. 5** Terminal Neu5Ac contributes to meningococcus species specificity. **a** A *CMAH*-KO clone, in which endogenous CMAH was mutated on both alleles (causing reading frame shift) using a CRISPR/Cas9 approach (Supplementary Fig. 8), was derived from C166 mouse endothelial cells. Wild-type or CMAH-KO C166 cells expressing exogenous YFP-tagged human $\beta_2$AR and CEACAM-1 (to induce mock bacterial adhesion) were infected with piliated non-capsulated Opa+ (2C43 SiaD- Opa+) meningococci. Receptor distribution was examined by fluorescence microscopy. Receptor accumulated under bacterial colonies (arrows) in CMAH-KO but not in wild-type cells, reflecting the requirement of Neu5Ac (instead of Neu5Gc) for pili interaction with the $\beta_2$AR. Left panel: representative images, bar: 10 μm. Right panel: recruitment of $\beta_2$AR was quantified and expressed as the % of bacterial colonies that recruited transfected $\beta_2$AR-YFP. Mean values ± SEM. Statistical significance was calculated by t test, **$p < 0.01$. **b** To examine the reconstitution of a typical $\beta_2$AR-dependent signaling pathway induced by meningococcus in the CMAH-KO C166 clone, cells were transfected with a plasmid encoding human CEACAM-1, pre-incubated or not with isoproterenol for 1 h (to induce the endocytosis of endogenous $\beta_2$AR), and then exposed to piliated non-capsulated Opa+ (2C43 SiaD− Opa+) meningococci. Ezrin recruitment under colonies was used as a marker of $\beta_2$AR signaling. ***$P < 0.001$ (ANOVA, $n = 3$ in duplicate). Representative images are shown in the left panel, bar: 10 μm

subsequent experiments, adherent meningococci were submitted to orbital rotation aimed at applying a centrifugal force on bacteria. Rotation significantly enhanced ezrin accumulation under bacterial colonies both in wt and ΔT meningococci. However, the effect was significantly larger for wt colonies, indicating that maximal signaling depends on the additive effect of PilT-induced pilus retraction and exogenous forces applied to bacteria. Iso-proterenol pre-treatment inhibited meningococcus signaling both under static conditions and after orbital agitation.

The potential mechano-transducing role of the $\beta_2$AR revealed in context of meningococcal infection was further investigated using Neu5Ac-selective WGA-coated beads in a reconstituted cell system. Beads were added to HEK-293 cell monolayers for 1 h, to promote WGA binding to Neu5Ac, and then submitted to orbital rotation. In the absence of rotation, only basal translocation of ezrin and $\beta_2$AR was observed under the beads lying on cells expressing exogenous $\beta_2$AR and myc-tagged β-arrestins. Rotation appliance significantly enhanced ezrin accumulation, but no accumulation was obtained in cells expressing the AT1R-YFP as receptor, or in cells expressing $\beta_2$AR alone (Fig. 7). Thus, traction forces specifically applied on the $\beta_2$AR via its glycans, even in the absence of bacteria, can induce β-arrestin-selective activation and downstream ezrin recruitment. In contrast to bacterial-mediated activation, however, the requirement for tandem glycan chains at a specific distance was not observed for the activation via WGA-coated beads. Indeed, wt $\beta_2$AR or $\beta_2$AR mutants lacking either N-glycan chain (N6A or N15A) were activated to a similar extent.

## Discussion

From our data it appears that the activation of the $\beta_2$AR by meningococcal Tfp involves traction forces applied via the bacterial pili on glycan chains located in the N-terminal extracellular region of the receptor. Part of the forces is generated intrinsically by the bacterial PilT ATPase, which controls Tfp retraction, extrinsic shear stress forces created by blood flow, and applied on bacteria providing the additional mechanical signal. Although previous studies reported that host–pathogen interactions can involve host cell glycans[31] and that signaling properties of some receptors can be modulated by glycosylation[32,33], the GPCR activation mechanism reported here is unique. It involves NANAs terminating two glycan chains located at a specific distance in the receptor N-terminus, independently of the protein backbone. Interestingly, the amino-acid residues mediating the activation of $\beta_2$AR signaling have been identified for the major pilus component PilE[34]. The PilE C-terminus domain, which contains a disulfide-bonded region (D-region), was found to be critical for inducing activation of the host cell response. In particular, a hyper-variable region confers specificity for signaling in endothelial cells via the $\beta_2$AR. Among the four residues that are critical for signaling, two are positively charged lysine residues, an observation that fits well with the fact that this region would interact with the negatively charged NANAs terminating the $\beta_2$AR glycan chains. The particular distance of the receptor glycan chains required for bacterial-induced signaling activation probably reflects structural constraints, such as pilin orientation or

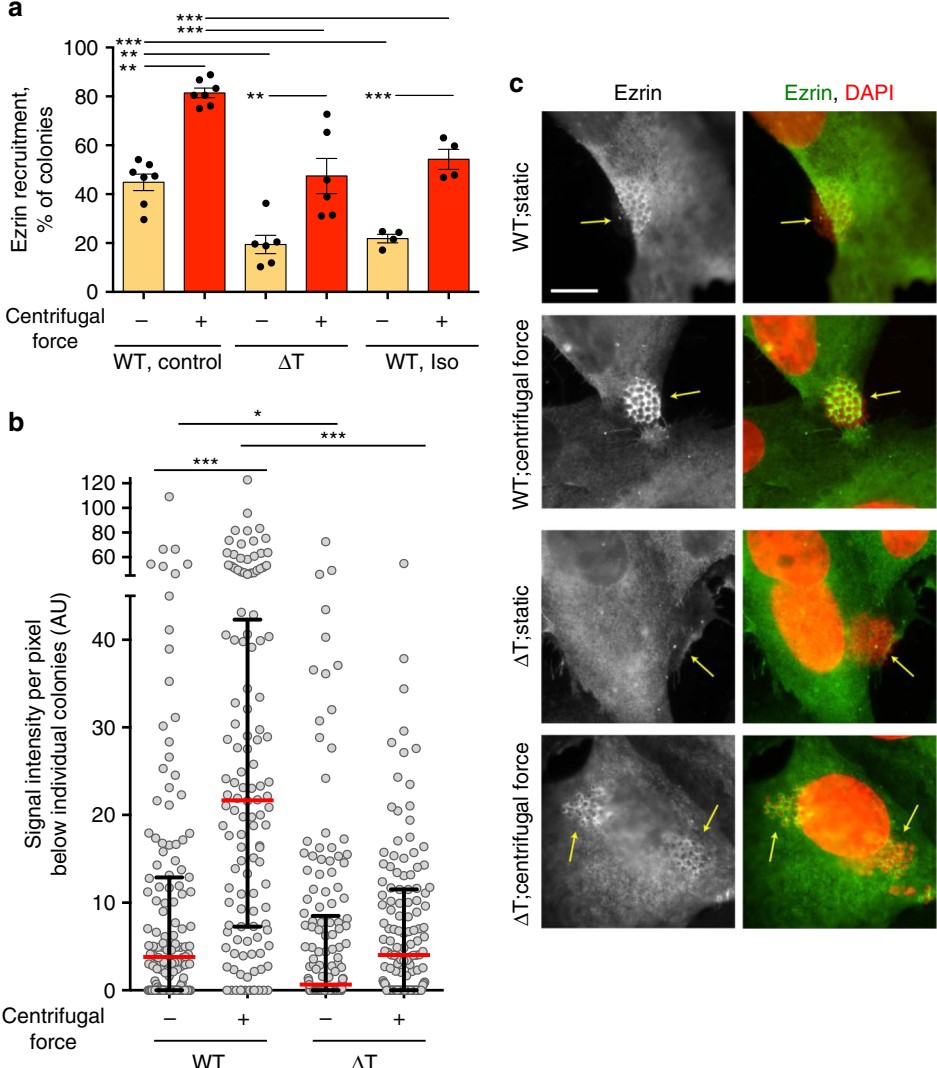

**Fig. 6** Pili- and environment-generated forces concur to the $\beta_2AR$ activation. **a** Endothelial hCMEC/D3 cells were infected with wild-type or $\Delta T$ N. *meningitidis* and submitted or not to centrifugal forces with or without isoproterenol pre-incubation. A particular attention was paid to ensure that under "static condition" cells were submitted to minimal shear stress (see Methods). Cells were then processed for the analysis of ezrin recruitment under bacterial colonies as in Fig. 3. Histograms correspond to mean values ± SEM. Data were analyzed by ANOVA with Dunnett's multiple comparisons test. ***$P < 0.001$; **$p < 0.01$. **b** The intensity of ezrin recruitment was also quantified for each colony individually (see Methods) and expressed as arbitrary intensity per pixel below colonies. Each dot represents one colony. Median values (in red) with interquartile ranges are shown. A Kruskal–Wallis test was used to determine significant differences between median values. ***$P < 0.001$; *$p < 0.05$. **c** Representative images, bar: 10 µm

spacing within the bacterial pilus. Supporting this hypothesis is the cooperativity that exists between the pilin subunits for full receptor activation (Supplementary Fig. 9).

The $\beta_2AR$ is not the only GPCR displaying two glycan chains in its N-terminus at the "meningococcus infection-permissive" distance of 8–9 residues, since this feature is found in roughly 10% of GPCRs. However, only few of them are expressed in endothelial cells (i.e., bradykinin B2R, angiotensin AT2R, acetylcholine M3R) and they are not necessarily pre-associated with the adhesion receptor CD147 as the $\beta_2AR$, which is a key parameter in terms of meningococcal pathophysiology[11]. Possibly because of the limited number of receptors that are potentially activated by pili, the recruitment of ezrin in response to mechanically stimulated $\beta_2AR$ might be restricted to cell types expressing high concentrations of $\beta$-arrestins, such as endothelial cells. Indeed, a robust recruitment of ezrin under meningococcal colonies[9], and to WGA-coated beads in the HEK-293-reconstituted cell system, required the expression of exogenous $\beta$-arrestin, while these cells

contain endogenous $\beta$-arrestins. Although $\beta_2AR$ is accountable for most of the meningococcus-induced signaling in endothelial cells, a residual 10–15% ezrin recruitment under bacterial colonies was still observed upon receptor KO in hCMEC/D3 cells, consistent with the existence of some minor $\beta_2AR$-independent signaling.

Recent structural and molecular studies have markedly expanded our comprehension of the molecular mechanisms, which control GPCR function[35]. For class A GPCRs (the class to which the $\beta_2AR$ belongs to) that are activated by non-peptide ligands, the orthosteric binding site, through which endogenous agonists activate the GPCR, is generally located in the middle of the seven-transmembrane helical bundle, between the extracellular loops and the middle plane of the plasma membrane. Allosteric ligands modulate receptor binding and activity by binding to sites, distinct from the orthosteric site, which can be located in variable positions. The binding site for a "smallmolecule" allosteric modulator of the $\beta_2AR$ was localized, for

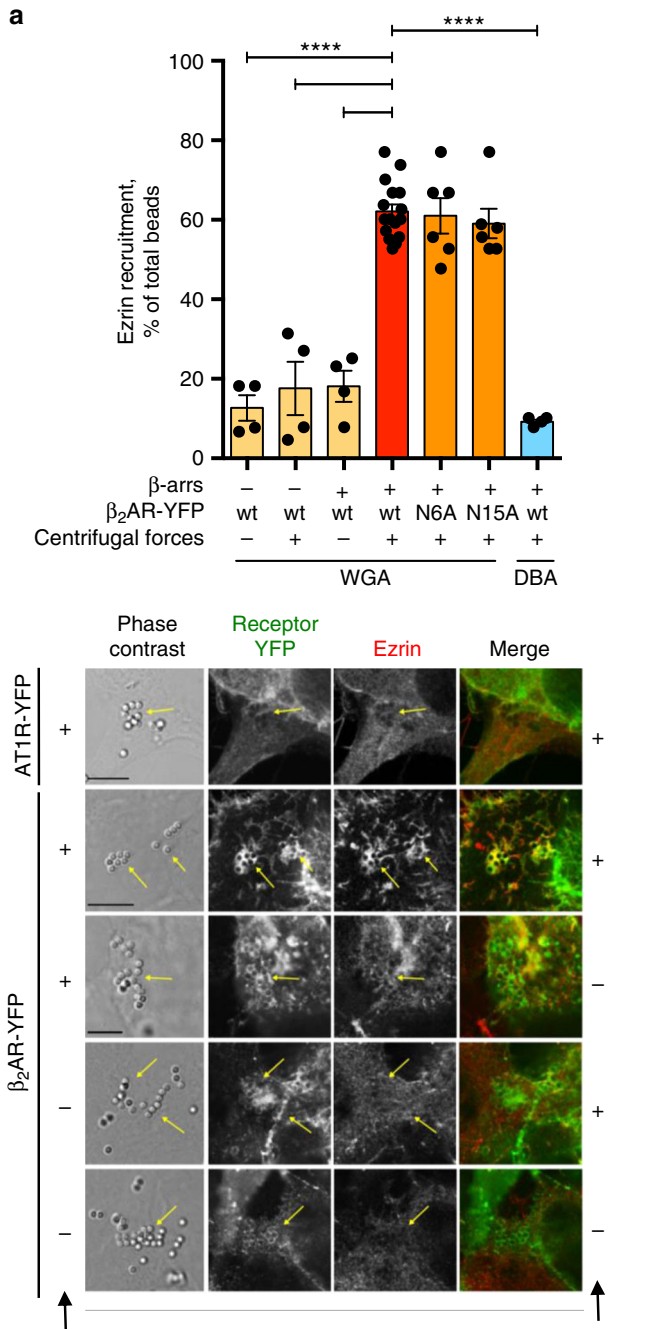

**Fig. 7** Tractions forces applied via lectin-coated beads activate the $\beta_2$AR. **a** Neutravidin-agarose beads (1 μm) were coated with biotinylated WGA or control dolichos biflorus agglutinin (DBA) as described in Methods. The latter lectin has specificity for α-linked *N*-acetylgalactosamine. They were then let to sediment for 1 h on HEK-293 cells expressing the indicated YFP-tagged wt of mutant receptors, with or without exogenous $\beta$-arrestins ($\beta$-arrs), in 2 cm$^2$ well plates before 15 min of orbital rotation at 100 r.p.m. Cells were fixed and labeled with anti-ezrin antibodies. Ezrin recruitment under beads was used as a marker of receptor signaling. ****$P < 0.0001$, ANOVA. **b** Representative images of experiments in **a** are shown, bar: 10 μm. The AT1R was used as a negative control

example, in a pocket formed by the cytoplasmic ends of transmembrane segments 1, 2, 6, and 7, the intracellular loop 1, and the helix 8[36]. In contrast, an allosteric modulator of the M2 muscarinic receptor was positioned directly above the orthosteric

binding site, engaged in extensive interactions with the extracellular vestibule, including the second extracellular loop[37]. A recent article identified the exosite for the aryloxyalkyl tail of salmeterol, a selective $\beta_2$AR agonist with long-lasting action. Interestingly, the aryloxyalkyl tail of salmeterol extends toward the extracellular surface of the receptor, occupying a cleft formed by residues from extracellular loops 2 and 3 and the extracellular ends of the transmembrane segments 6 and 7[38]. These observations, together with the fact that the orthosteric site for peptide ligands can involve the N-terminal domain and is stabilized by extracellular loops[39], suggest a mechanism for the pilus-mediated activation of the $\beta_2$AR, whereby the traction exerted via bacterial pili bound to glycan chains engages the helical bundle region, via the mobilization of the N-terminal region, which establishes interactions with the transmembrane core. This model is supported by the observation that applying pulling forces on the $\beta_2$AR, via beads coated with lectins specific for the exposed NANA terminating the glycan chains of the receptor, reproduces a pili-like activation.

An interesting issue is whether the mechanical activation of the $\beta_2$AR would be affected by the simultaneous presence of circulating catecholamines in vivo. Experiments were performed in which isoproterenol and bacteria were added simultaneously to cells in culture (Supplementary Fig. 10). This treatment markedly inhibited the recruitment of both receptor and ezrin under the colonies, while the isoproterenol-promoted receptor endocytosis was maintained in these cells, probably explained by the fact that not all receptors are engaged by bacterial pili in a given cell and by the use of saturating concentrations of the agonist that target all available receptors. The pathophysiological significance of these findings is however difficult to appreciate, since in vivo, catecholamine concentration is usually below the $k_D$ for the $\beta_2$AR, with only a small fraction of receptors bound and potentially internalized. Noteworthy, in case of meningococcus infection with sepsis, catecholamine infusion is part of the recommended supportive treatment; in addition to its action on cardiovascular parameters, it might limit the infection by enhancing receptor endocytosis.

The ligand-independent mechanical activation of other class A GPCRs in endothelial cells (and also cardiomyocytes and fibroblasts) was reported in response to fluid shear[40], osmotic stress [40,41], or to uniaxial traction[42]. GPCR coupling to $G_{q/11}$ was found to be involved in signal transduction of several mechanically activated receptors in the context of myogenic vasoconstriction[43]. However, reminiscent of our findings, it was also reported that mechanical stretch can induce $\beta$-arrestin-selective signaling downstream of AT1Rs, the conformational change in $\beta$-arrestin induced by the stretch being similar to that induced by a $\beta$-arrestin-biased ligand[44]. The mechano-transducing properties of the $\beta_2$AR reported here indicate potential additional functions for this receptor in the vascular system. For example, macrophages, which adhere to and roll over endothelial cells before penetrating into tissues, express the NANA-specific sialoadhesin Siglec-1[31], which share the same glycan specificity as Mal-I. SIglec-1 binding to endothelial cell $\beta_2$AR might induce some signaling in the context of cell-to-cell interactions and/or diapedesis.

*Neisseria meningitidis* is a strict human pathogen. In addition to factors, which contribute to the survival of bacteria in the blood, such as transferrin binding or resistance to the complement, our data indicate that the mechanism of meningococcus-induced signaling itself contributes to species selectivity. Indeed, the presence of a single additional oxygen atom in Neu5Gc, the predominant mammalian form of NANA, compared to Neu5Ac, the predominant human form, is sufficient to impede pili binding to $\beta_2$AR glycans and subsequent Tfp-mediated mechanical activation of the receptor. Our data suggest that additional factors

might also contribute to meningococcal selectivity for its host. Indeed, mouse endothelial cell expressing Neu5Ac following inactivation of CMAH showed impaired initial adhesion. This observation indicates that the adhesion receptor (CD147 in humans) might also contribute to pathogen selectivity.

The *Alu*-mediated loss of CMAH about 2–3 million years ago, which determines the absence of Neu5Gc in humans and its replacement by Neu5Ac in the glycocalyx was probably maintained in humans because of some selective advantage, such as the resistance to an ancestral Neu5Gc-binding pathogen related to *P. reichenowi*, which infects non-human primates[45]. Nevertheless, this early adaptation caused several pathophysiological consequences in humans, in addition to the susceptibility to meningococcal infection reported here. For example, the major erythrocyte-binding antigen-175 of *Plasmodium falciparum*, the parasite causing the most severe forms of human malaria, apparently evolved to take selective advantage of the excess of the Neu5Ac on human erythrocytes[46]. Also, *Salmonella typhi*, an exclusive human pathogen that causes typhoid fever, produces a pathogenic toxin. This toxin binds to and is toxic for cells expressing glycans terminated in Neu5Ac over glycans terminated in Neu5Gc[47].

In conclusion, the $\beta_2$AR activation mechanism by meningococcus, the specificity of this pathogen for humans, and the tether for meningococcal Tfp anchoring to endothelial cells actually implicate a common molecular element, Neu5Ac, a NANA form that is only found in humans. Traction forces applied via Tfp on Neu5Ac-terminated glycans of the $\beta_2$AR activate the receptor. In addition to unraveling a mechanism contributing to meningococcal species selectivity, these findings represent the first example of a glycan-dependent mode of allosteric mechanical activation of a GPCR.

## Methods

**Cell lines**. The hCMEC/D3 human cell line, which retains main features of primary brain endothelial cells and express endogenous $\beta$-arrestins, $\beta_2$AR, and CEACAM-1, was described previously[48] (a kind gift from Pierre-Olivier Couraud, Institut Cochin, Paris, France). hCMEC/D3 cells were grown at a density of 25 × $10^3$ cells per cm$^2$ in flasks coated with 5 µg/cm$^2$ rat tail collagen type I (BD Bioscience), in EBM-2 medium (Lonza) supplemented with 5% of fetal calf serum (FCS), 1% of penicillin–streptomycin, 1.4 µM hydrocortisone, 5 µg/ml ascorbic acid, 10 mM HEPES, and 1 ng/ml basic fibroblast growth factor at 37 °C in a humidified incubator in 5% $CO_2$.

HEK-293 (ATCC CRL-1573) cells were grown in flasks coated with 0.01% poly-L-lysine (Sigma) in Dulbecco's modified Eagle's medium (DMEM) supplemented with 10% FCS and 1% penicillin–streptomycin in 5% $CO_2$-enriched humidified atmosphere.

The mouse endothelial cell line C166 (ATCC CRL-2581), which express endogenous $\beta_2$AR, was grown in DMEM supplemented with 10% FCS and 1% penicillin–streptomycin in 5% $CO_2$-enriched humidified atmosphere.

The BHK cell line (ATCC PTA-3544) was grown in DMEM supplemented with 10% FCS and 1% penicillin–treptomycin in 5% $CO_2$-enriched humidified atmosphere.

**Bacterial strains**. Serogroup C meningococcal strain 8013, designated as 2C43[49], the non-encapsulated Opa expressing derivative, designated as SiaD-2C43[8], and a mutant deficient in PilT activity derivative designated $\Delta$T[30] were used as specified in the Methods section (own collection). Bacteria were grown on Gonococcal Broth (GCB) agar plate with Kellogg's supplement and the appropriate antibiotics. Kanamycin was used at 100 µg/ml and chloramphenicol was used at 6 µg/ml. Other strains have been used either variant of PilE pilin (namely, PilESA and PilESB) or deleted of *pilV* gene ($\Delta$PilV) in the context of PilESA or PilESB variants, or deleted of *pilE* gene ($\Delta$PilE)[5,10,50].

**Treatment of cells and infections**. On the day of infection, bacteria grown from an overnight culture on GCB agar plate were adjusted to OD$_{600}$ = 0.05 and incubated for 2 h at 37 °C in pre-warmed cell culture medium. Bacteria were added to cells at a multiplicity of infection of 100 and allowed to adhere for 30 min. Unbound bacteria were washed with fresh medium and infection was allowed to proceed for 2 h.

HCMEC/D3 cells were infected with wt 2C43 strain or its $\Delta$T derivative (Figs. 3, 5, and 6). HEK-293 and C166 cells were infected with the non-encapsulated SiaD-2C43 strain and its derivatives: $\Delta$PilV, $\Delta$PilE and PilESA, or PilESB variants.

Lectins used in infection-inhibition experiments were purchased from Vector laboratories (the complete list is shown in Supplementary Table 1 and used at a final concentration of 20 µg/ml, as recommended by the manufacturer). They were added 1 h before infection and again after 30 min of infection when fresh medium was added. Whereas MAL-I did not inhibit bacterial adhesion to host cells, WGA partially inhibited bacterial adhesion at the used concentration. Since we showed previously that signaling was independent of adhesion, ezrin recruitment was examined below adherent colonies.

To quantify the role of pilus retraction and/or that of external forces like shear stress on ezrin recruitment, we set up a particular protocol of infection. To achieve stringent static conditions, cells were not washed after the 2 h infection and 16% paraformaldehyde (PFA) was added gently, directly to the infection medium (final PFA concentration: 4%). To apply continuous centrifugal forces mimicking shear stress, cells were placed on an orbital shaker at 100 r.p.m. Fixation was performed by adding 16% PFA on top of the cells.

Potassium depletion experiments were performed as described[8]. Briefly, cells were rinsed with K$^+$-free buffer (140 mM NaCl, 20 mM HEPES, 1 mM CaCl$_2$, 1 mM MgCl$_2$, 1 mg/ml D-glucose, pH 7.4) and then incubated in hypotonic buffer (K$^+$-free buffer diluted 1:1 with distilled water) for 5 min. Cells were then washed three times in the K$^+$-free buffer and incubated for 20 min at 37 °C in the same buffer. In order to maintain proper bacterial growth during the potassium depletion assay, cell culture medium was added to K$^+$-free buffer (1:5) during meningococcal infection.

For sialyl-transferase inhibition experiments, various concentrations of the sialyl-transferase inhibitor 3FP-Neu5Ac (Merck) were added to cell culture medium during 3 weeks prior to the experiments. At high concentrations ($\geq$300 µM), 3FP-Neu5Ac drastically inhibited adhesion of meningococci to endothelial cells. Consequently, lower concentrations of the drug were used (100 and 200 µM), which were compatible with adhesion, although markedly inhibiting ezrin recruitment.

**Flow experiments**. Cells were grown on disposable flow chambers (Slide VI, IBIDI$^{TM}$) coated with 5 µg/cm$^2$ rat tail collagen type I. In total, $10^7$ bacteria were added on top of the cells and allowed to adhere for 15 min without flow. Subsequently, a flow corresponding to a shear stress of 0.4 dyn/cm$^2$ was applied using a syringe pump (Harvard Apparatus). To quantify the surface of colonization, cells and bacteria were fixed using 4% PFA during 20 min; bacteria were then stained using anti-2C43 rabbit antibodies (1:1000, generously provided by Dr. P Mangeat CNRS, UMR5539, Montpellier, France) and cells were stained using phalloidin. Bacteria were analyzed using an Incucyte S3 analysis system (Sartorius). Four images were acquired per IBIDI channel (×20 magnification). Surface of colonizing bacteria was determined using the Incucyte S3 software and expressed as median of colonization index with interquartile range (total surface of bacteria in µm$^2$ over total surface occupied by cells in µm$^2$).

**Immunofluorescence on *Nm*-infected cells**. After infection, cells were washed three times and fixed for 20 min in PBS containing 4% PFA and permeabilized for 5 min in PBS containing 0.1% Triton X-100. Cells were incubated in PBS containing 0.1% bovine serum albumin (BSA) for 30 min and then with polyclonal anti-ezrin rabbit antibody (1:1000) in PBS containing 0.1% BSA for 1 h. After three washings in the same buffer, DAPI (4′,6-diamidino-2-phenylindole; 0.5 µg/ml) was added to Alexa-conjugated secondary antibodies for 1 h (A-11003 and A-11035, Thermo Fisher Scientific, 1:200). After additional washings, coverslips were mounted in Mowiol. Image acquisition was performed on a laser-scanning confocal microscope (Leica TCS SP5). Images were collected and processed using the Leica Application Suite AF lite (Leica) software.

**Expression and purification of pilin recombinant proteins**. Recombinant His-tag pilin fusion proteins were produced as perisplam-directed proteins as described before[51]. Fragments of *pilE*, *pilV*, and *comP* genes lacking the region coding for the amino-acid residues 1 to 28 of the full-length proteins were amplified by PCR from the existing plasmids pmal-PilE, pmal-PilV, and pmal-ComP[9] and subcloned in pET22b (Novagen) between *Bam*HI and *Xho*I restriction sites. The primer sequences used are provided in Supplementary Table 2.

The pET22b plasmid carries a signal peptide to direct the recombinant protein into *Escherichia coli* periplasm, where disulfide bond formation can occur. These plasmids were transformed in the *E. coli* BL21 (DE3) (Thermo Fisher Scientific) strain and grown at 30 °C for 3 h in lysogeny broth (LB), and 16 h at 16 °C in LB + 1 mM isopropyl $\beta$-D-1-thiogalactopyranoside. The fusion proteins were extracted from the periplasm and loaded onto a Ni-NTA column (Thermo Scientific). Bound His-pilin were eluted using elution buffer (50 mM NaH$_2$PO$_4$, pH 8, 300 mM NaCl, 250 mM imidazole) and then dialyzed using an Amicon 10k device and resuspended in PBS at a concentration of 1 mg/ml. Purity and quality of recombinant proteins were assessed by sodium dodecyl sulfate-polyacrylamide gel electrophoresis (SDS-PAGE) and Coomassie blue staining.

**Plasmids and transfections**. hCMEC/D3 and C166 cells were transfected using the Amaxa Nucleofactor Kit for HUVEC (Amaxa Biosystem), according to the manufacturer's instructions, with 2 μg of plasmids per $10^6$ cells. HEK-293 cells were transfected using 1 μl of Lipofectamine 2000 (Invitrogen) and 0.1 μg of each plasmid per 100 μl of final mix. BHK cells were transfected using GeneJuice Transfection Reagent (Novagen) according to the manufacturer's instructions. Cells ($10 \times 10^6$) were transfected with plasmid DNA containing HA-EC1-TM1$\beta_2$AR-YFP (5 μg) and processed with HTRF 44 h later. The plasmid encoding HA-EC1-TM1$\beta_2$AR-YFP was constructed using a two-step procedure. First, the YFP cDNA (pEYFP-N1 vector; Clontech) was amplified by PCR using primers containing *Bst*EII site. The PCR product was then subcloned in phase downstream of the codon for V67 in the HA-$\beta_2$AR-YFP plasmid[52] using the *Bst*EII site. The plasmid was sequenced to verify the phase and the absence of undesired mutations (DNA sequence facility, Institut Cochin).

Construction of plasmids coding for myc-$\beta$arr1/2 and GFP-$\beta$arr2[53], the human $\beta_2$AR fused to YFP ($\beta_2$AR-YFP)[54], the AT1R[55] were described previously. The plasmid encoding CEACAM-1[56] was kindly provided by Dr. Mumtaz Virji (Department of Cellular and Molecular Medicine, School of Medical Sciences, University of Bristol, Bristol, UK). The plasmid encoding the mouse $\beta_2$AR fused to YFP ($\beta_2$AR-YFP) was obtained by PCR amplification from mouse $\beta_2$AR cDNA and subsequent sub-cloning of the PCR product into the pEYFP-N1 plasmid (Clonetech) using *Xho*I and *Hin*dIII restriction sites. The primer sequences used ($\beta_2$AR_Fwd and $\beta_2$AR_Rv) are provided in Supplementary Table 2.

The plasmid coding for $\beta_2$AR-N6A-YFP, $\beta_2$AR-N15A-YFP, HA-AT1R-1Ngly$\beta_2$AR, and HA-AT1R-2Ngly$\beta_2$AR (+6 to +11 mutant) were obtained by site-directed mutagenesis of the $\beta_2$AR-YFP or the AT1R plasmids. The HA-$\beta_2$AR-(N + 11) construct, consisting of a HA-tagged $\beta_2$AR mutant in which two alanine residues were introduced between the leucine residue at position 12 and the alanine at position 13 (relative to the wt $\beta_2$AR sequence), and the HA-$\beta_2$AR-N6A and $\beta_2$AR-N15A constructs were obtained by site-directed mutagenesis (QuickChange®, Stratgene).

**Homogeneous time-resolved FRET**. HTRF measures energy transfer of between cryptate donors (610HATAA, monoclonal antibody (Mab) anti HA-Tb cryptate, Cisbio) and acceptor fluorophores (61HISDLA, Mab anti-6HIS-d2, Cisbio) with low background in intact cells. After transfection with the appropriate HA-tagged or control constructs, HEK-293 cells or mouse BHK cells were seeded in white 96-well assay plate (Costar) at $1.5 \times 10^5$ cells/well in 200 μl of DMEM supplemented with 10% fetal bovine serum. Twenty-four hours later, cells were rinsed with 100 μl of Tris/Krebs buffer (20 mM Tris-Cl (pH 7.4), 118 mM NaCl, 5.6 mM glucose, 1.2 mM $KH_2PO_4$, 1.2 mM $MgSO_4$, 4.7 mM KCl, and 1.8 mM $CaCl_2$) supplemented with 0.1% bovine serum albumin. Cells were then incubated in 100 μl of the same buffer with recombinant pilins (Pil: PilV, PilE, or ComP at 5 μM final concentration) containing a N-terminal 6×His-tag during 60 min at 4 °C. The anti-HA monoclonal antibody coupled with Lumi4-Terbium cryptate and the anti-6His monoclonal antibody labeled with a fluorescence acceptor molecule (d2, Cisbio, Inc.) were then added (both at 1:1000 dilution) for 60 min (room temperature (RT)) before excitation at 320 nm and reading (FRET emission at 620 and 665 nm) in a Mithras2 LB 943 HTRF reader. Owing to the high sensitivity and to the low concentration of antibodies required, the signal could be measured directly after incubation without washing out the unbound antibody. A 400 μs reading was used after a 100 μs delay to allow for decay of short-lived endogenous fluorescence signals. Fluorescence at 620 nm corresponds to the total europium cryptate signal, whereas fluorescence at 665 nm is the FRET signal. The ratio $R = (F_{665\,nm}/F_{620\,nm}) \times 10^4$ was computed. The FRET signal was expressed as arbitrary unit (AU), calculated using the following equation: HTRF signal AU $= (R - R_{neg})/(R_{neg})$, where $R_{neg}$ is the ratio for the negative energy transfer control (non-transfected cells incubated with the FRET-donor antibody)[57].

**CRISPR/Cas9 genome editing**. The ADRB2-KO cell line was obtained from human hCMEC/D3 cells using pSpCas9(BB)-2A-GFP (PX458) plasmid (a gift from Dr. Feng Zhang: Addgene plasmid #48138) and the specific guide sequence (5′-TGACGTCGTGGTCCGGCGCA-3′). HCMEC/D3 cells were transfected with the constructed plasmids (1 μg/100 μl) using AMAXA Transfection System (Lonza), HUVEC Transfection Kit (Lonza), and program HUVEC-OLD. The day after, cells were trypsinized and suspended in cold Hank's balanced salt solution/FCS (2%). GFP-positive cells were selected using an Aria 409 III Cell Sorter (BD Bioscience). Cells were seeded at decreasing densities in 6-well plates and grown to isolate proliferative clones. To select KO clones, genomic DNA was extracted using the PureLink Genomic DNA Purification Kit (Invitrogen), following the manufacturer's protocol, and the genomic region surrounding the CRISPR-targeted site was amplified by PCR. Products were purified using PCR Product Purification (Invitrogen), following the manufacturer's protocol, and then analyzed by Sanger sequencing. KO clones carrying frame-shift mutations and/or large deletion were selected and amplified.

The CMAH-KO cell line was obtained from mouse C166 cells, using lentiCRISPRv2 (a gift from Dr. Feng Zhang: Addgene plasmid #52961[58]) and the specific guide sequence (5′-TTGAGGTTGGCAACTTCAGC-3′). The plasmid was transduced in C166 cells using third-generation lentiviruses, and puromycin-resistant cells were selected by adding puromycin (4 μg/ml) to the cell culture

media 48 h after transfection. Cells were then seeded at decreasing densities in 6-well plates. Clones were selected after 2 weeks and amplified before selection.

KO clones were selected as described above.

**Agarose-Neutravidin beads and incubation with cells**. Before experiment, 20 μl Neutravidin beads (FluoSphere™ NeutrAvidin™, Invitrogen) of 1 μm diameter were incubated with 20 μg of appropriate biotinylated lectins (VectorLabs) in 50 μl OPTIMEM (Invitrogen) for 30 min at RT and washed twice in 500 μl OPTIMEM following centrifugation at 16,000 r.p.m. for 2 min. Beads were resuspended in 50 μl OPTIMEM. One hour before experiments, transfected HEK-293 cells were washed twice with warm OPTIMEM and incubated in serum-free OPTIMEM in 24-well plates containing 500 μl OPTIMEM per well. Ten microliters of coated beads were then added per well. Beads were incubated for 1 h without any agitation. Cells were then stirred on an orbital plate for 15 min at 100 r.p.m. and supplemented with 8% PFA under continuous agitation. In control wells (without agitation), 8% PFA was added drop-wise to avoid fluid displacement. (In each case, the final concentration of PFA was 4%.) Fixed cells were washed and processed for ezrin immunolabeling.

**Quantification and statistical analysis**. Statistical analyses were performed with GraphPad Prism. Multiple comparison analyses were assessed with a non-parametric one-way Kruskal–Wallis test and Bonferroni's multiple camparison test and Dunnett's multiple comparisons test. Non-parametric two-tailed Mann–Whitney U-test was performed on data with only two groups. P values are reported in the figures and figure legends where significant. Error bars are shown as standard error of the mean (SEM) and are indicated in the figures.

To quantify the recruitment of ezrin or that of $\beta$-arrestin-GFP under bacterial colonies, cells were first treated for immunofluorescence as described above. Then, each coverslip was analyzed under a wide-field fluorescence microscope equipped with a ×100 objective, a ultraviolet filter, a 488-nm short-pass filter and a 546-nm long-pass filter. For each field, the number of colonies was counted and the presence or absence of the protein of interest below colonies was determined. The focal plane was adjusted just above the apical membrane. Recruitment of the protein of interest was counted as positive if: (i) the recruited protein was accumulated below colonies in a ring shape around the bacteria. This shape is also called honeycomb shape, and (ii) if the fluorescence intensity was greater than the general fluorescence at the plasma membrane. At least 50 colonies were observed per coverslip. Each experiment was repeated several times in triplicate. Data were examined for significance using the Prism GraphPad software.

Quantification of ezrin recruitment below colonies was also performed by measuring the intensity of fluorescence signals (as in Fig. 6b): 10 fields per slides were acquired with a Zeiss Apotome 2.0. Each colony was analyzed using the ImageJ software. Briefly, the DAPI layer and ezrin staining layer were extracted and a threshold mask was applied on the ezrin staining layer to suppress the background staining (normal ezrin staining at the plasma membrane) and to transform the layer into a black and white image. The mean value of pixel intensity below colonies reflects the total area covered by ezrin staining and is expressed as Arbitrary Intensity of ezrin recruitment.

**Trypsin shaving and immunoprecipitation of $\beta_2$AR**. HEK-239 cells were transfected with the indicated plasmids and cultivated for 36 h. The cells were harvested in PBS-EDTA (0.5 μM EDTA in PBS) and digested with trypsin (1 μg/ml) (Promega, catalog #V511A) according to the manufacturer's protocol at 37 °C for 1 h. Following the trypsin digestion, the cell pellets were lysed in lysis buffer (50 mM HEPES, pH 7.4, 250 mM NaCl, 2 mM EDTA, 0.5% NP-40, 10% glycerol, and complete protease inhibitor (Roche)) on ice for 30 min. Cell lysates were then centrifuged at 13,000 r.p.m. for 15 min at 4 °C. Supernatants were transferred to a fresh tube and the presence of cleaved receptor N-termini was verified by western blot. Proteins were separated on SDS-PAGE on a 10% (w/v) polyacrylamide resolving gel. The proteins were transferred to nitrocellulose membranes. Membranes were blocked with 5% skimmed milk in 0.1% TBST (Tris-buffered saline containing 0.1% Tween-20 (Sigma)) for 1 h and at RT. The membranes were incubated with primary antibodies anti-HA (Cell Signaling) (1:1000) and anti-vinculin (1:5000) (Sigma) overnight at 4 °C. The following day, the membranes are washed three times with 0.1% TBST and incubated with secondary antibodies IRDye® 800 CW Donkey anti-Rabbit and IRDye® 680 CW Donkey anti-Mouse (LI-COR) (1:10,000). The blots were washed two times with 0.1% TBST and one time with TBS and were developed using Odyssey® CLx (LI-COR).

The trypsin-digested supernatant was immunoprecipitated using anti-HA beads (EZview™ Red HA Agarose beads, Sigma) overnight at 4 °C. The beads were washed three times with PBS and further subjected to MS analysis or NANA detection.

**MS analysis of HA-purified peptides**. Trypsin-cleaved N-termini of wt and mutant $\beta_2$AR bound to HA beads were submitted to On-Bead PNGase digestion (0.5 μl, Fast PNGase F, New England Biolabs) in a final volume of 15 μl of PNGase F denaturing and reducing buffer for 10 min at 50 °C on a thermomix (Eppendorf). Supernatants were then collected and diluted to a 50 μl final volume with 50 mM $(NH_4)_2CO_3$

Worthington's TLCK (tosyl-L-lysyl-chloromethane hydrochloride)-treated chymotrypsin (125 ng) was used to digest the eluted deglycosylated N-terminal peptides in solution at 37 °C for 45 min. Resulting products were analyzed by nano liquid chromatography hyphenated to a Q-Exactive Plus MS (Thermo) operating in data-dependent scheme with an inclusion list containing $12 m/z$ for peptides of interest. Briefly, peptides loaded on the chromatography were trapped and washed with 0.1% trifluoroacetic acid and 2% acetonitrile (ACN) in milliQH$_2$O on a C$_{18}$ reverse-phase pre-column (Acclaim Pepmap 100 Å pores, 5 μm particles, 2 cm long, 75 μm inner diameter) for 3 min at 5 μl/min. Trapped peptides were then separated on a C$_{18}$ reverse-phase analytical column (2 μm particle size, 100 Å pore size, 75 μm inner diameter, 25 cm length) with a 1.5 h gradient starting from 99% of solvent A containing 0.1% formic acid in milliQH$_2$O and ending in 40% of solvent B containing 80% ACN and 0.085% formic acid in milliQH$_2$O. The MS acquired data throughout the elution process. The MS scans spanned from 400 to 2000 Th with AGC target $1 \times 10^6$ with 100 ms Maximum Ion Injection Time (MIIT) and resolution of 70,000. Higher energy collisional dissociation (HCD) fragmentations were performed on the 10 most abundant ions with a dynamic exclusion time of 30 s. Precursor selection window was set at 4 Th. HCD normalized collision energy was set at 30% and MS/MS scan resolution was set at 17,500 with AGC target $1 \times 10^5$ within 100 ms MIIT. Spectra were recorded in profile mode. MS data were analyzed using Mascot 2.5.1 (www.matrixscience.com). The database used was a concatenation of human sequences from the Uniprot-Swissprot database (release 2019-02, 20492 sequences) and a customized database of 70 modified sequences including the four ADRB2-mutated HA-tagged sequences (WT, N6A, +11, and N15A), and a list of in-house and common contaminant sequences (292 sequences). Cysteine carbamidomethylation was set as constant modification; asparagine deamidation was set as variable modification to allow detection of peptide modified by PNGase F enzymatic deglycosylation.

**Sialic acid detection assay.** NANA quantification on immunoprecipitated wt and mutant β$_2$AR N-termini after trypsin digestion was performed using the NANA Colorimetric/Fluorometric Assay Kit (Bio Vision Incorporated, Catalog #K566-100). Free NANA was released from the HA-bead immunoprecipitated material by heating the sample with 2 M acetic acid at 80 °C for 3 h, followed by the addition of 2 M NaOH to neutralize the solution. The standard curve and the concentration of the NANA for the different samples were determined according to the manufacturer's protocol. To normalize the results according to the actual amount of fragments immunoprecipitated with HA beads, 10% of the beads were set aside and eluted with 2× loading dye, and analyzed by western blot. The specific anti-HA-labeled material was quantified using Image Studio Lite (LI-COR).

**Reporting summary.** Further information on research design is available in the Nature Research Reporting Summary linked to this article.

## Data availability

Data supporting the findings of this manuscript are available from the corresponding authors upon reasonable request. A reporting summary for this article is available as a Supplementary Information file.

The source data underlying Figs. 1b–d, 2a, b, 3a, c, 3e, 3g, 5a, b, 6a, b, 7a and Supplementary Figs. 1a–d, 4c, 5b, 6b, 9 are provided as a Source Data file.

The MS proteomics data have been deposited to the ProteomeXchange Consortium via the PRIDE partner repository with the dataset identifier PXD015389.

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

## Acknowledgements

We thank N. Goudin of the Necker Institute Imaging Facility, S. Fabrega, L. Nay, and S. Hadj Hamou of the Viral Vectors and Genes Transfer facility (VVTG) for their technical support. This work was supported by ANR-14-IFEC-0006-01, ANR-15-CE15-0002, and ANR MECAD-R 2019 grants. S.M. and X.N. are supported by INSERM, CNRS, Université Paris Descartes, and the Fondation pour la Recherche Médicale.

## Author contributions

M.C., S.D., and S.M. designed the experiments. S.M and M.C. wrote the manuscript. S.D. performed HTRF, S.M. molecular biology experiments, M.C. and Z.V. cell biology experiments, M.L., C.B., and M.C. mechano-transduction assays, M.C. K.S., and R.M.D. performed glycan analysis, and K.S. and F.G. performed proteomic analysis. X.N., P.M.R., and C.R.-M. discussed data and hypotheses and reviewed the manuscript.

## Competing interests

The authors declare no competing interests.
