## [Peer Review File · Nature Communications]

Reviewers' Comments:

Reviewer #1:

Remarks to the Author:

This well written paper uses elegant methods to demonstrate the importance of Neu5Ac on N-glycans of beta2-adrenergic receptor (b2AR) in mediating interactions with meningococcal type IV pili (Tfp). Importantly, only Neu5Ac (made by humans, who cannot convert Neu5Ac to Neu5Gc because of inactivation of the enzyme CMAH), but not Neu5Gc (expressed by non-human primates and lower animals) is important for interactions with Tfp. This important advance adds to our understanding of the human specificity of meningococcal infection. My only suggestion is to provide some direct evidence for the respective nonulosonates (either Neu5Ac or Neu5Gc) on the glycans at the two critical N residues on b2AR, as detailed below.

1. Related to the data in Fig. 2b, where spacing of the glycans was considered critical and was subsequently presumed to be Neu5Ac in Fig. 3 based on lectin inhibition, it would be important to show that both glycan residues in question are sialylated. Altering the distance between the N residues could impact glycosylation (and sialylation), therefore it is important to formally demonstrate Neu5Ac at both positions on the +9 ('wild-type' b2AR) cell line and on one of the mutant cells with fewer (eg, +6) or more (eg, +11) amino acids between the N residues.

2. Similarly, for Fig. 4a, expression of Neu5Gc on b2AR should be confirmed on the mouse C166 cell line. Lower animals make both Neu5Ac and Neu5Gc and there is variation in the relative amounts of each of the nonulosonates expressed in different tissues (eg, Malykh et al, Glyconj J. 1998: 15(9):885-93). The CMAH deletion that is accompanied by increased b2AR recruitment strongly suggests that Neu5Ac is involved. Demonstrating that both N residues in the wild-type cells express Neu5Gc that are replaced by Neu5Ac when CMAH is inactivated will provide solid structural data to back the conclusions.

3. Page 6, 4th line from bottom – add 'the' between 'Since' and 'meningococcus'.

Reviewer #2:

Remarks to the Author:

In the current manuscript Virion et al., illustrate the molecular mechanism by which the meningococcus can infect human.

The involvement of the β 2AR was already known, as well as the importance of the nanonewton forces generated by the filamentous structures (Tfp) for the infection of the bacteria.

The novelty of this work is the demonstration that the two asparagine-branched glycan chains present at the N-termini of the β 2AR at a particular distance from each other are necessary and sufficient to support receptor activation. Specifically they found out that terminally exposed sialic acids in the β 2AR are essential for both receptor activation by pathogens, explaining the basis for species selectivity.

Besides finding the target on the receptor for the pathogens, they explain the molecular mechanism employed by the them. They show that the Tfp structure binds to the N-termini of the β 2AR and applying their forces to two glycan chains can in turn activate the β 2AR that will function as a mechano-sensor receptor leading to stabilization of the bacteria colonies.

The paper is overall well written and logic. As a novel approach they used a FRET based approach for determine the binding of the component of the Tfp to the β 2AR. They use as a read out of the infection the Ezrin recruitment. However some additional control and clarification could improve the paper.

Major

1. The authors to demonstrate that the PiIV and PiIE interact with the N-terminus of the β 2AR using a FRET based approach (figure 1b)

They state that the GFP fluorescence was used as a control for the receptor expression level however they do not show this control anywhere. I think will be important to include this control since change in the expression level of the receptor could impact the HTRF signals. Additionally , they show only one time point of the binding (HTRF signal after 60 minutes incubation), would it be possible to look at the kinetics of this process.

2. The authors state that "a dose dependent specific HTRF was measured with recombinant His-PiIV and PiIE" however in fig 1c they only show it for PiIV , I suggest to add the data for PiIE or to remove it from the text. Could you also put this concentration in a physiological context? What are the concentration expected in vivo ?

3. The authors should show the YFP signals related to the experiments in figure 1d so that the reader could actually see that the receptor is expressed equally.

4. The author show that two asparagine-branched glycan chains are necessary and sufficient to support receptor activation.

5. In figure 6 the author show that traction forces could induce ezrin recruitment " via its glycan" however this is not directly demonstrated. I think such an experiment expressing instead β 2AR-N6A or β 2AR-15A could directly prove it.

Minor

1. I found fig 1b a bit complicated to interpret, I suggest to separate the HEK from the BHK findings since they anyway confer different messages.

2. HEK endogenously express β - arrestins, however in all the experiments beta arrestins are always transfected. Are the basal concentration of these protein not enough?

Reviewer #3:

Remarks to the Author:

This manuscript by Virion et al focuses on investigating mechanisms of *N. meningitidis* interaction with endothelial cells and its effects on the β 2 adrenergic receptor. The authors propose that mechanical force applied to the glycans on the amino terminus of the β 2 receptor leads to activation signaling through the β arrestin pathway.

This manuscript is quite difficult to evaluate due to use of unconventional assays throughout, and the complete lack of a standard signaling assay. This work also relies heavily on prior literature from the same group, which further complicates interpretation. Overall I believe claims made by the authors are insufficiently supported by the data presented.

Most importantly, the authors repeatedly make very strong claims regarding an essential role for β 2 adrenergic receptor in *N. meningitidis* entry (beginning with the very first sentence of the abstract). These claims are not adequately supported either in the present manuscript or in prior work referenced therein. More specifically, claims regarding biased signaling through the arrestin pathway would require an operational model analysis of signaling, based on a comparison of cAMP dose-response curves with β -arrestin dose-response curves (Kenakin and Christopoulos, *Nat Rev Drug Discov* 2013). No such assays are included in this manuscript, and these experiments do not appear to have been reported elsewhere previously. Without a proper pharmacological analysis claims regarding biased signaling are unsupported and should be removed.

In fact, claims regarding signaling of any kind through the $\beta 2$ receptor appear to be unsupported. Radioligand competition binding assays, cAMP assays, and arrestin recruitment assays are all standard and widely used techniques for analyzing GPCR signaling. Why are all of these absent here? The ezrin recruitment/organization assay is an unconventional readout for adrenergic signaling, and it should at the very least be accompanied by more standard assays. Without conventional signaling readouts it is difficult to conclude anything regarding whether *N. meningitidis* affects $\beta 2$ receptor signaling at all. Claims regarding a direct interaction between the $\beta 2$ N-terminus and *N. meningitidis* pilins are also incompletely supported. A biochemical measure of the binding is required for claims regarding a direct interaction (for example, surface plasmon resonance or even just a pull-down assay using purified pilins and $\beta 2$ receptor amino terminus).

A smaller concern is that the authors make several claims about $\beta 2$ adrenergic receptor dimers. It is unclear that $\beta 2$ receptors actually dimerize in any physiologically meaningful way, and it is well established that dimerization is not required for the receptor to show normal functional properties (Whorton MR et al PNAS 2007; Rasmussen et al Nature 2011).

Reviewer #4:

Remarks to the Author:

The studies presented in the current manuscript describe the ability of gram negative bacterium *Meningococcus* to adhere to the endothelial cells. Such an adherence initiates cellular signaling leading to stable establishment of the virulent colonies which over the period of time can translocate to through the endothelial barrier. In this context, the investigators show that type IV pili of the bacteria can activate beta2-adrenergic receptor which is known to complex with the adhesion receptor CD147. Mechanistically, the study shows that the pili interacts with the exposed N-terminal glycan chains and exerts traction forces sufficient to mediate beta2-adrenergic receptor activation. The beta2-adrenergic receptor activation is identified by the staining of the receptor along with the overlay with ezrin using confocal microscopy. Although the studies are well thought out, some of the experiments needs to provide better proof of the conclusions by using more rigorous set of experimental tools.

a) The HTRF studies are interesting as they have used a single transmembrane modified segment of beta2-adrenergic receptor with GFP tag to assess for the binding of pilins. However, a key missing piece in this data is the evidence that this partial single transmembrane receptor representing protein is expressed on the plasma membrane. The reviewer agrees that is a powerful tool but was disappointed that the authors did not provide a comparison on binding affinity of pilins to the single transmembrane receptor vs. the seven transmembrane beta2-adrenergic receptor. This would have helped in easy interpretation of the HTRF.

b) More importantly, no data is presented to show that this single transmembrane protein mimics some the structure features of the seven transmembrane beta2-adrenergic receptor. This is important, given the observation that pilin binds to the receptor through structural determinants and not the amino-acid sequence. Also, did the control GFP have a transmembrane segment but without the beta2-adrenergic sequence?

c) Furthermore, given that beta2-adrenergic receptor forms a complex with CD147, the membrane receptor that engages with meningococcus, it would be critical for the authors to show that this single membrane partial receptor can function similarly when co-expressed with CD147 which would have strengthened the findings.

d) The authors show that meningococcus activates beta2-adrenergic receptor by using the glycan chain on the N-terminal extracellular domain identifying that the distance between the chains is

the primary determinant for activation. Given that binding to the CD147 on the cells determines the adherence, it would be important to show whether co-expression of CD147 will change the sensitivity of beta2-adrenergic activation and thus in turn ezrin recruitment. Furthermore, whether there is an additive effect of beta2-adrenergic activation by using a combination of both the pilins i.e., PilV and PilE together. In addition, it will greatly strengthen the manuscript, if the authors can identify the amino acids on PilV and PilE that mediates the activation of beta2-adrenergic receptor.

e) The authors state the beta2-adrenergic receptor activates beta-arrestin pathway following meningococcus treatment but have not shown any data supporting this idea/statement which can be easily done through beta-arrestin recruitment assays. Is beta-arrestin activation required for ezrin recruitment to the inner membrane of cells subjacent to the colony on the cells? This is important as ezrin activation is used as a surrogate for beta2-adrenergic receptor activation following meningococcus treatment.

f) It is very exciting to observe that inhibition of sialyl-transferases with the 3Fp-Neu5Ac leads to reduced ezrin recruitment despite the presence of meningococcus indicating the specificity of the glycan in receptor activation. A key question is whether beta-arrestin recruitment is also inhibited or not? While the authors indicate that pilin binding activates the receptor, it would be important to show what happens in the context of beta2-adrenergic receptor endocytosis. They mention that ISO pretreatment may reduce the number of receptor and thus the ezrin recruitment but given the exciting data on pilin mediated activation, it would be important to assess whether simultaneous combination i.e., pilin + ISO will have differential effects on the beta-arrestin and/or ezrin recruitment. This may be an important aspect to study as in vivo, the endothelial cells would be in communication/contact with both beta-adrenergic receptor agonists as well as the bacteria.

g) Finally, the studies on the PilT deficient cells is very interesting as it indicates that the nanonewton forces generated by the pili retraction could activate the receptor. Such an idea is supported by the centrifugal experiments However the key underlying principle is that ISO leads to receptor internalization thereby reducing the available receptors for pilin interaction but have not provided as an quantitative measure of amount of beta-adrenergic receptor that would be sufficient to mediate colony formation/ezrin recruitment as data in Fig 5A indicates that despite loss of beta-receptors from cell surface, ezrin is still recruited potentially suggesting receptor independent mechanisms. The authors will need to elaborate this observation as this mechanism seems to be as important as the beta2-adrenergic receptor mediated ezrin recruitment.

Minor:

Typo in line 19 on page 4 should be "pili-coated" instead of "pili-coaed".

Reviewer #1 (Remarks to the Author):

This well written paper uses elegant methods to demonstrate the importance of Neu5Ac on N-glycans of beta2-adrenergic receptor (b2AR) in mediating interactions with meningococcal type IV pili (Tfp). Importantly, only Neu5Ac (made by humans, who cannot convert Neu5Ac to Neu5Gc because of inactivation of the enzyme CMAH), but not Neu5Gc (expressed by non-human primates and lower animals) is important for interactions with Tfp. This important advance adds to our understanding of the human specificity of meningococcal infection. My only suggestion is to provide some direct evidence for the respective nonulosonates (either Neu5Ac or Neu5Gc) on the glycans at the two critical N residues on b2AR, as detailed below.

We thank the reviewer for the overall appreciation of our work.

1. Related to the data in Fig. 2b, where spacing of the glycans was considered critical and was subsequently presumed to be Neu5Ac in Fig. 3 based on lectin inhibition, it would be important to show that both glycan residues in question are sialylated. Altering the distance between the N residues could impact glycosylation (and sialylation), therefore it is important to formally demonstrate Neu5Ac at both positions on the +9 ('wild-type' b2AR) cell line and on one of the mutant cells with fewer (eg, +6) or more (eg, +11) amino acids between the N residues.

To formally demonstrate the presence of sialic acid residues terminating glycan chains of the β 2AR N-terminus we used the following stepwise strategy (illustrated in the **Supplementary Fig 7**). The N-terminus of the β 2AR contains an Arg residue preceding the first TM. This residue can be targeted by trypsin digestion to release almost the entire N-terminus. Full-length HA-tagged wt or mutant β 2ARs expressed in intact HEK-293 cells were submitted to trypsin digestion. The expected wt 36 amino acid residue fragments liberated in the supernatant and IP with anti-HA-coated beads displayed an apparent MW of about 20-30KDa due to its N-glycosylation (the expected MW of the peptide is about 4kDa). The N-termini from N6A and N15A β 2AR migrated faster than the fragments from wt and (+11) β 2AR since they contain only one N-linked glycan chain (**Fig 4a,b**). These fragments were analyzed by LC-mass spectrometry after additional PNGase F and chymotrypsin digestion to confirm that they correspond to the expected N-termini (**Fig 4c**).

After enrichment on anti-HA-coated beads, part of the samples was heated in acidic conditions to release Neu5Ac from the glycan chains, and liberated sialic acid was quantified by a colorimetric assay (**Fig 4d**). Sialic acid released from wt and (+11) β 2AR N-termini was equivalent and significantly higher than that released from N6A and N15A β 2AR. These data strongly support the conclusion that both glycan chains at position 6 and 15 can individually undergo terminal sialylation and that both glycan chains from wt and (+11) β 2AR are similarly sialylated.

2. Similarly, for Fig. 4a, expression of Neu5Gc on b2AR should be confirmed on the mouse C166 cell line. Lower animals make both Neu5Ac and Neu5Gc and there is variation in the relative amounts of each of the nonulosonates expressed in different tissues (eg, Malykh et al, Glyconj J. 1998: 15(9):885-93). The CMAH deletion that is accompanied by increased β 2AR recruitment strongly suggests that Neu5Ac is involved. Demonstrating that both N residues in the wild-type cells express Neu5Gc that are replaced by Neu5Ac when CMAH is inactivated will provide solid structural data to back the conclusions.

The reviewer comment is valid: as in animals Neu5Ac is the substrate of CMAH and Neu5Gc the product, variable proportions of each form can be found in different tissues. Since the transfection of mouse C166 endothelial cells is not comparable to what can be obtained in HEK293 cells in terms of level of expression and homogeneity around 5% of transfected cells versus 90%), we used a different approach to demonstrate the enrichment in Neu5Ac in the CMAH-KO clone. Early studies reported that WGA is selective for Neu5Ac compared to Neu5Gc (Bhavanandan VP and Katlic AW, J. Biol. Chem. 254, 4000-4008, 1979; Monsigny M et al, Eur. J. Biochem. 104, 147-153 1980; Maget-Dana R et al, Eur. J. Biochem. 114, 11 -16, 1981). We performed precipitation assays with WGA-coated agarose beads from extracts of C166 wt and C166-CMAH KO cells expressing exogenous β 2AR-GFP,

with equivalent counts of fluorescent (effectively transfected) cells. Blots were probed with anti-GFP antibodies. The glycosylated form of β 2AR-GFP was substantially more abundant in precipitations from C166-CMAH KO cells providing indirect evidence that these cells contain a higher proportion of the Neu5Ac form than parental C166 (Supplementary Fig 8e).

3. Page 6, 4th line from bottom – add ‘the’ between ‘Since’ and ‘meningococcus’.

Corrected

Reviewer#2

In the current manuscript Virion et al., illustrate the molecular mechanism by which the meningococcus can infect humans.

The involvement of the β 2AR was already known, as well as the importance of the nanonewton forces generated by the filamentous structures (Tfp) for the infection of the bacteria.

The novelty of this work is the demonstration that the two asparagine-branched glycan chains present at the N-termini of the β 2AR at a particular distance from each other are necessary and sufficient to support receptor activation. Specifically they found out that terminally exposed sialic acids in the β 2AR are essential for both receptor activation by pathogens, explaining the basis for species selectivity.

Besides finding the target on the receptor for the pathogens, they explain the molecular mechanism employed by them. They show that the Tfp structure binds to the N-termini of the β 2AR and applying their forces to two glycan chains can in turn activate the β 2AR that will function as a mechano-sensor receptor leading to stabilization of the bacteria colonies.

The paper is overall well written and logic. As a novel approach they used a FRET based approach for determine the binding of the component of the Tfp to the β 2AR. They use as a read out of the infection the Ezrin recruitment. However some additional control and clarification could improve the paper.

We thank the reviewer for the overall appreciation of our work.

Major

1. The authors to demonstrate that the PilV and PilE interact with the N-terminus of the β 2AR using a FRET based approach (figure 1b).

They state that the GFP fluorescence was used as a control for the receptor expression level however they do not show this control anywhere. I think will be important to include this control since change in the expression level of the receptor could impact the HTRF signals.

The fluorescence values of HTRF experiments have been included (new Supplementary Fig. 1a) together with all requested additional controls related to Figure 1.

Additionally, they show only one time point of the binding (HTRF signal after 60 minutes incubation)

The duration of 60 minutes was chosen because in preliminary experiments 60 min corresponded to the beginning of the binding plateau. These preliminary studies have been included in the Supplementary Fig. 1b.

2. The authors state that “a dose dependent specific HTRF was measured with recombinant His-PilV and PilE” however in fig 1c they only show it for PilV, I suggest to add the data for PilE or to remove it from the text

PilE was added in Fig 1c

Could you also put this concentration in a physiological context? What are the concentrations expected in vivo?

The HTRF experiment was conducted to confirm the hypothesis that the bacterial ligands (PilV and PilE) are capable of binding to the β 2AR N-terminus. The experimental conditions do not reflect at all what occurs *in vivo* when meningococcus interacts with endothelial cells: in a pathophysiological context, there are no free pilins but polymers of pilins arranged in bundles, which form the bacterial pilus. The avidity of immobilized ligand polymers, rather than affinity, better determines pilin binding to the receptors expressed in endothelial cells. Consistently, our previous studies on PilV and PilE binding to the adhesion receptor (CD147) indicated that the affinity of pilin monomers is actually weak and that polymerization compensates this by providing avidity (Bernard S et al, Nat. Med. 20, 725-731, 2014). We showed that purified pilins do activate β 2AR-dependent β -arrestin recruitment *in vitro* only if pre-bound to antibody-coated fixed bacteria (staphylococci, Coureuil et al, Cell 2010 Fig 7). During meningococcal infection the number of bacteria in the blood is quite variable; in most

cases only a few bacteria crossing the BBB are sufficient to infect meninges and cause meningitis. In conclusion, what is likely critical for infection *in vivo* is that a colony of piliated bacteria provides a sufficient high density of pilin polymers in proximity to CD147- β 2AR complexes at the luminal surface of endothelial cells.

3. The authors should show the YFP signals related to the experiments in figure 1d so that the reader could actually see that the receptor is expressed equally.

Western Blots probed with the anti-GFP antibody (in the new **Supplementary Fig. 1e**) show the expression of human or mouse β 2AR-YFP in HEK-293 cells. The level of transfection in C166 endothelial cells is much lower (about 5% of transfected cells) compared to HEK-293 cells. The C166 cells expressing β 2AR-YFP were identified and counted visually under the fluorescence microscope as shown, for example **in Fig. 5**. For the calculation of signaling colonies only colonies above fluorescent cells were counted. Note that in C166 cells meningococcal colonies accumulate near fluorescent, β 2AR-YFP-expressing cells, which also express the mock-adhesion receptor CEACAM-1. In the absence of CEACAM there is no adhesion. In the revised version of the manuscript these points are specified in the legend of the **Supplementary Fig. 1e**.

4. The authors show that two asparagine-branched glycan chains are necessary and sufficient to support receptor activation.

No question in this sentence

5. In figure 6 the author show that traction forces could induce ezrin recruitment “via its glycan” however this is not directly demonstrated. I think such an experiments expressing instead β 2AR-N6A or β 2AR-15A could directly prove it.

In the new Figure 6 (bacterial-promoted traction, former Figure 5), the distance constraint observed for meningococcal-induced signaling probably depends on structural constraints, such as pilin orientation or spacing within the bacterial pilus (as mentioned in the discussion section). The WGA-bead-promoted activation does not have the same limitation. This is demonstrated by an experiment comparing the signal induced by the centrifugal forces on cells expressing β 2AR wt, β 2AR-N6A or β 2AR-15A (added into the new **Figure 7**). The best argument that “*traction forces could induce ezrin recruitment via its glycan*”, comes from control experiments we had with beads coated with lectins that do not interact with sialic acid. Under these conditions no signal is elicited upon rotation of culture plates (**Figure 7**), whereas WGA-beads, which recognize sialic acid) elicit a signal.

Minor

1. I found fig 1b a bit complicated to interpret, I suggest to separate the HEK from the BHK findings since they anyway confer different messages.

The panel has been split in the revised version as requested

2. HEK endogenously express β -arrestins, however in all the experiments beta arrestins are always transfected. Are the basal concentration of these protein not enough?

In our early studies (Coureuil et al 2010) we reported that endogenous β -arrestin is translocated to bacterial colonies in endothelial cells. However the level of β -arrestin expression is variable in different cell types (it is not very high in HEK-293 cells) and the signal to background ratio may be weak in some microscopy experiments where it is necessary to count positive signals, particularly when exogenous receptors are used. For this reason in all experiments performed in reconstituted cell systems exogenous β -arrestin is added.

Reviewer #3 (Remarks to the Author):

This manuscript by Virion et al focuses on investigating mechanisms of N. meningitidis interaction with endothelial cells and its effects on the β_2 adrenergic receptor. The authors propose that mechanical force applied to the glycans on the amino terminus of the β_2 receptor leads to activation signaling through the β arrestin pathway.

This manuscript is quite difficult to evaluate due to use of unconventional assays throughout, and the complete lack of a standard signaling assay. This work also relies heavily on prior literature from the same group, which further complicates interpretation. Overall I believe claims made by the authors are insufficiently supported by the data presented.

Most importantly, the authors repeatedly make very strong claims regarding an essential role for β_2 adrenergic receptor in N. meningitidis entry (beginning with the very first sentence of the abstract). These claims are not adequately supported either in the present manuscript or in prior work referenced therein. More specifically, claims regarding biased signaling through the arrestin pathway would require an operational model analysis of signaling, based on a comparison of cAMP dose-response curves with β -arrestin dose-response curves (Kenakin and Christopoulos, Nat Rev Drug Discov 2013). No such assays are included in this manuscript, and these experiments do not appear to have been reported elsewhere previously. Without a proper pharmacological analysis claims regarding biased signaling are unsupported and should be removed.

In fact, claims regarding signaling of any kind through the β_2 receptor appear to be unsupported. Radioligand competition binding assays, cAMP assays, and arrestin recruitment assays are all standard and widely used techniques for analyzing GPCR signaling. Why are all of these absent here? The ezrin recruitment/organization assay is an unconventional readout for adrenergic signaling, and it should at the very least be accompanied by more standard assays. Without conventional signaling readouts it is difficult to conclude anything regarding whether N. meningitidis affects β_2 receptor signaling at all. Claims regarding a direct interaction between the β_2 N-terminus and N. meningitidis pilins are also incompletely supported. A biochemical measure of the binding is required for claims regarding a direct interaction (for example, surface plasmon resonance or even just a pull-down assay using purified pilins and β_2 receptor amino terminus).

General reply:

We agree with the reviewer that the entire story of meningococcus interaction with the β_2 AR in the context of infectious meningitis that we have been investigating for about ten years is totally unconventional compared with the classical paradigm of ligand-receptor interaction. In the present case there is no free soluble ligand but rather a pathogen that interacts with host cell receptors via insoluble ligands covalently attached to bacterial appendages. Thus the classical pharmacological approaches based on saturation curves at the equilibrium, competition assays, second messenger assays etc... used for analyzing GPCR signaling are not appropriate here. We have developed instead approaches that are commonly used to investigate signaling phenomena dependent on cell-to-cell interaction.

Specific answers

“Most importantly, the authors repeatedly make very strong claims regarding an essential role for β_2 adrenergic receptor in N. meningitides entry (beginning with the very first sentence of the abstract). These claims are not adequately supported either in the present manuscript or in prior work referenced therein”

We previously reported (Coureuil et al, Cell 2010) that the β_2 AR is absolutely necessary (siRNA experiments, agonist-promoted receptor endocytosis) for the induction of the cellular protrusions, which stabilize the colonies under flow and for the opening of endothelial cell intercellular spaces. We could reconstitute a meningococcus-promoted signaling in HEK-293 cells by simply expressing exogenous β_2 AR and β -arrestins (Cell paper Fig 3).

To answer to a specific point raised by Reviwer#4 we additionally show in the revised version of the manuscript that meningococcus-induced signaling is dampened to minimal levels in human endothelial cells, in which the β_2 AR is knocked-out by a CRISPR-Cas9 approach (Supplementary Fig 4). In conclusion, we are convinced that the role of the β_2 AR in the meningococcal interaction with endothelial cells has been demonstrated beyond any reasonable doubt.

« More specifically, claims regarding biased signaling through the arrestin pathway would require an operational model analysis of signaling, based on a comparison of cAMP dose-response curves with β -arrestin dose-response curves... No such assays are included in this manuscript, and these experiments do not appear to have been reported elsewhere previously »

We previously showed that cAMP production is NOT induced in endothelial cells by the pathogen (Cell paper, supplemental fig 2) contrasting with β -arrestin recruitment under colonies and β -arrestin-mediated Src activation. These findings justify the concept that meningococcus induces a β -arrestin-biased activation of the β 2AR.

“These claims are not adequately supported either in the present manuscript or in prior work referenced therein”..... “Without conventional signaling readouts it is difficult to conclude anything regarding whether N. meningitidis affects β 2 receptor signaling at all”

This statement is surprising to us since deciphering step by step the signaling pathways elicited by meningococcus via the β 2AR in endothelial cells was precisely the major objective of our Cell paper.

We reported that the pathogen induces the recruitment of (endogenous and exogenous) β -arrestins under colonies and that this phenomenon is β 2AR and GRK2 dependent (Cell paper, Fig.1 and 2).

We reported the key role of the β -arrestin-dependent activation of Src for recruiting actin in cell protrusions and of the β -arrestin-dependent recruitment of junctional proteins under bacterial colonies (Cell paper, Fig. 6). Confirming previous findings, to address a point raised by Reviewer #4 (to document the effect of 3FP-Neu5Ac on β -arrestin recruitment), β arr2-GFP recruitment under a colony is shown in the new Fig 3d-e.

“Radioligand competition binding assays”

Meningococcal pilins interact with an allosteric binding site distinct from the classical binding pocket for agonists and antagonists. We reported in the Cell paper (Figure 2E) that saturating propranolol does not inhibit meningococcal signaling. The symmetrical experiment cannot be done since meningococci engage a minimal fraction of β 2ARs expressed by endothelial cells in our experimental conditions.

“The ezrin recruitment/organization assay”

The demonstration that the recruitment of cytoskeletal proteins (actin or ezrin) represents the end point of the β 2AR- β -arrestin biased activation by meningococcus has been fully documented and analyzed step by step in our Cell paper (Coureuil et al, 2010) and corroborated by the information provided by other studies (Maïssa, 2017). Directly connecting β 2AR- β -arrestin signaling events with downstream ezrin or actin recruitment, we reported that inhibition of β 2AR or GRK2 or β -arrestins or Src (the various elements of the signaling cascade) all inhibited this recruitment.

A smaller concern is that the authors make several claims about β 2 adrenergic receptor dimers. It is unclear that β 2 receptors actually dimerize in any physiologically meaningful way, and it is well established that dimerization is not required for the receptor to show normal functional properties (Whorton MR et al PNAS 2007; Rasmussen et al Nature 2011).

We agree that the functional significance of GPCR dimerization is still a matter of debate. It was not our objective to open this debate in the context of the present story. Close proximity of exogenous β 2AR protomers (<50 nm, a distance compatible with homo-dimerization) has been reported *in vitro* using different approaches by many groups (including us). We used this proximity just once as a positive control for our HTRF study in Fig 1b.

Reviewer #4 (Remarks to the Author):

The studies presented in the current manuscript describe the ability of gram-negative bacterium *Meningococcus* to adhere to the endothelial cells. Such an adherence initiates cellular signaling leading to stable establishment of the virulent colonies which over the period of time can translocate to through the endothelial barrier. In this context, the investigators show that type IV pili of the bacteria can activate beta2-adrenergic receptor, which is known to complex with the adhesion receptor CD147. Mechanistically, the study shows that the pili interacts with the exposed N-terminal glycan chains and exerts traction forces sufficient to mediate beta2-adrenergic receptor activation. The beta2-adrenergic receptor activation is identified by the staining of the receptor along with the overlay with erzrin using confocal microscopy. Although the studies are well thought out, some of the experiments needs to provide better proof of the conclusions by using more rigorous set of experimental tools.

We thank the reviewer for the overall appreciation of our work.

a) The HTRF studies are interesting as they have used a single transmembrane modified segment of beta2-adrenergic receptor with GFP tag to assess for the binding of pilins. However, a key missing piece in this data is the evidence that this partial single transmembrane receptor representing protein is expressed on the plasma membrane. The reviewer agrees that is a powerful tool but was disappointed that the authors did not provide a comparison on binding affinity of pilins to the single transmembrane receptor vs. the seven transmembrane beta2-adrenergic receptor. This would have helped in easy interpretation of the HTRF.

The initial set up of the HTRF experiment was performed using full-length 7TM β 2AR-GFP. This experiment has been added in the **Supplementary Fig 1 (panel d)** and shows that the HTRF signal can be obtained with full-length receptor as well. However, the interpretation of these data in terms of affinity is complex. The HTRF assay was used in the attempt to demonstrate that the basic “bacterial ligand” can interact in some way with the receptor N-terminus. Even though the single TM is correctly located at the plasma membrane (the HA-tag involved in the HTRF assay is indeed extracellular, since it is recognized by the antibody in non-pemeabilized intact cells) and the HTRF data quite reproducible, the system only partially reflects the mechanism of interaction between the meningococcus and the receptor. The main point (see also our reply to reviewer #2, point 2: “...put this concentration in a physiological context”) is that in real life pilins are never under soluble monomeric form. Purified pilins can activate β 2AR-dependent β -arrestin recruitment only if pre-bound to antibody-coated fixed bacteria (Coureuil et al, Cell 2010 Fig 7). In the pathophysiological context, the interaction with the receptor is more driven by the avidity of polymeric bundles of pilins within bacterial pili than by the pilin monomer affinity (which is probably quite low).

b) More importantly, no data is presented to show that this single transmembrane protein mimics some the structure features of the seven transmembrane beta2-adrenergic receptor. This is important, given the observation that pilin binds to the receptor through structural determinants and not the amino-acid sequence. Also, did the control GFP have a transmembrane segment but without the beta2-adrenergic sequence?

As discussed in the previous point, a similar HTRF signal was obtained with the 7TM β 2AR. Based of what is known from structural data, the β 2AR N-terminus is probably flexible and unstructured in live cells. In addition, the 3-4 kDa size of the protein moiety of the N-terminus is small compared to the 20-30 kDa of the glycan moiety. It seems therefore unlikely that a particular structure of the peptide present in full length-receptor and important for bacterial binding would be missed in the single TM construct. The experiments shown in fig 2 support this hypothesis: the TM and the N-ter from the AT1R are competent for full length receptor activation by meningococcus provided that two glycan chains are present in the N-terminus at a particular distance. In our proposed model, the mechanical traction via pili-bound Neu5Ac is the key parameter for receptor activation. This mode of activation does not necessarily imply a particular structure of the receptor N-terminus. The GFP control was without the 1st TM.

c) Furthermore, given that beta2-adrenergic receptor forms a complex with CD147, the membrane receptor that engages with meningococcus, it would be critical for the authors to show that this single membrane partial receptor can function similarly when co-expressed with CD147, which would have strengthened the findings.

We have performed the experiment suggested by the Reviewer to compare the HTRF signal in the presence or absence of CD147 (Supplementary Fig.1c). In cells controlled for equivalent expression of the single TM, the presence of CD147 does not change significantly the signal compared to the control.

d) The authors show that meningococcus activates beta2-adrenergic receptor by using the glycan chain on the N-terminal extracellular domain identifying that the distance between the chains is the primary determinant for activation. Given that binding to the CD147 on the cells determines the adherence, it would be important to show whether co-expression of CD147 will change the sensitivity of beta2-adrenergic activation and thus in turn ezrin recruitment.

This comment is the extension “*in cellulo*” of the comment above. It is true that in endothelial cells the β 2AR activated by meningococcus is pre-associated to CD147. This pre-association is critical to provide to the pathogen a signaling-competent receptor near its adhesion site. However, we do not believe that CD17 plays a major role in the β 2AR activation process. Indeed, in reconstituted cells systems where adhesion is achieved via exogenous CEACAM-1 β 2AR is fully activated. To examine whether the presence of CD147 would more directly affect β 2AR signaling in intact cells, we have overexpressed CD147 in the HEK-293 reconstitution system where CD147-dependent adhesion is bypassed. We have selected a representative picture, in which 2 adjacent cells are visible, one expressing exogenous CD147 (right) and not the other (left) (data joined at the end of this file, Annex 1). Exogenous CD147 accumulates nicely under bacterial colonies together with ezrin, while the recruitment of ezrin is comparable in both cells. Thus whether or not exogenous CD147 is expressed, meningococci seem to activate β 2AR signaling to a similar extent.

Furthermore, whether there is an additive effect of beta2-adrenergic activation by using a combination of both the pilins i.e., PilV and PilE together. In addition, it will greatly strengthen the manuscript, if the authors can identify the amino acids on PilV and PilE that mediates the activation of beta2-adrenergic receptor.

A new experiment (Supplementary Fig. 9) shows the synergistic effect of PilE and PilV in the context of infectious bacteria. Briefly, some bacterial strains have been selected expressing either variant of PilE pilin (namely, PilESA and PilESB, the latter being the variant expressed by the strain 2C4.3 used for all other experiments), or deleted of *PilV* gene (Δ PilV) in the context of PilESA or PilESB variants, or deleted of *PilE* gene (Δ PilE). In the last case, due to the absence of the major PilE pilin, bacterial pili are totally absent. Either *PilV* gene deletion or PilESA variation caused a significant decrease of ezrin recruitment compared to standard PilESB strain. When these two modifications were combined the inhibition of ezrin recruitment was further enhanced (compare Δ PilV-PilESA with Δ PilV-PilESB).

The amino acid residues mediating the activation of the β 2AR have been identified for PilE in a previous study. The PilE C-terminus domain, which contains a disulfide-bonded region (D-region), is critical for the host cell response. In particular, a hyper-variable region confers specificity for signaling in endothelial cells via the β 2AR. Among the four residues which are critical for signaling, two are positively charged lysine residues, an observation that fits well with the fact that this region would interact with the negatively charged sialic acids terminating the β 2AR glycan chains (Miller et al, mBio 5: e01024-13, 2014). This information is included in the discussion of the revised manuscript. The situation is more complex for PilV, since the amino acid residues involved in signaling have not been clearly identified yet.

e) The authors state the beta2-adrenergic receptor activates beta-arrestin pathway following meningococcus treatment but have not shown any data supporting this idea/statement, which can be

easily done through beta-arrestin recruitment assays. Is beta-arrestin activation required for ezrin recruitment to the inner membrane of cells subjacent to the colony on the cells? This is important as ezrin activation is used as a surrogate for beta2-adrenergic receptor activation following meningococcus treatment.

Illustrating the connection between β -arrestin and ezrin recruitment, we have examined the β -arrestin recruitment in parallel to that of ezrin in endothelial cells treated with 3FP-Neu5Ac (new Fig 3d-e, to reply to the point “f” raised by the Reviewer, see below)

We would like to remind that the step-to-step elucidation of the β -arrestin-dependent pathway promoted by meningococcus was described in our article published in Cell in 2010 (a summary of the data is in the reply to Reviewer #3). In particular, it has been well established that the “end-point” ezrin recruitment is prevented by the inhibition of any of the upstream steps of activation, including β -arrestin recruitment.

f) It is very exciting to observe that inhibition of sialyl-transferases with the 3Fp-Neu5Ac leads to reduced ezrin recruitment despite the presence of meningococcus indicating the specificity of the glycan in receptor activation. A key question is whether beta-arrestin recruitment is also inhibited or not?

We have performed the experiment (new Fig 3d-e) and the data show that the inhibition of sialyl-transferases indeed prevents β -arrestin translocation beneath bacterial colonies.

While the authors indicate that pilin binding activates the receptor, it would be important to show what happens in the context of beta2-adrenergic receptor endocytosis. They mention that ISO pretreatment may reduce the number of receptor and thus the ezrin recruitment but given the exciting data on pilin mediated activation, it would be important to assess whether simultaneous combination ie., pilin + ISO will have differential effects on the beta-arrestin and/or ezrin recruitment. This may be an important aspect to study as in vivo, the endothelial cells would be in communication/contact with both beta-adrenergic receptor agonists as well as the bacteria.

We have done the experiment suggested by the Reviewer (data joined at the end of this file, Annex 2). Endothelial hCMEC/D3 cells were incubated with isoproterenol after infection: 15 min later the β 2AR accumulated in dots corresponding to endosomes while the amount of the receptor beneath the colonies decreased. In addition, in the HEK-293 reconstitution system, after 30 min incubation with ISO, ezrin recruitment was dramatically reduced compared to control cells (no ISO). These data indicate that some agonist-promoted receptor endocytosis is maintained in cells infected by the bacterium. Persistent endocytosis is probably explained by the fact that not all receptors are engaged by bacterial pili in a given cell and by the use of saturating concentrations of the agonist, which target all available receptors. The pathophysiological significance of these findings is however difficult to appreciate. *In vivo*, catecholamine concentration is usually below the K_D for the β 2AR and under these conditions only a fraction of receptors is bound and potentially internalized. Noteworthy, in case of meningococcus infection with sepsis, catecholamine infusion is part of the recommended supportive treatment in intensive care units; in addition to its action on cardiovascular parameters, it might limit the infection by enhancing receptor endocytosis.

g) Finally, the studies on the PilT deficient cells is very interesting as it indicates that the nanonewton forces generated by the pili retraction could activate the receptor. Such an idea is supported by the centrifugal experiments. However the key underlying principle is that ISO leads to receptor internalization thereby reducing the available receptors for pilin interaction but have not provided as an quantitative measure of amount of beta-adrenergic receptor that would be sufficient to mediate colony formation/ezrin recruitment as data in Fig 5A indicates that despite loss of beta-receptors from cell surface, ezrin is still recruited potentially suggesting receptor independent mechanisms. The authors will need to elaborate this observation as this mechanism seems to be as important as the beta2-adrenergic receptor mediated ezrin recruitment.

We agree with the Reviewer that data with both meningococci and lectin-coated beads display a 10-20% background, suggesting some potential β 2AR-independent mechanism. Also, it is difficult to quantitatively determine the number of receptor required for meningococci-induced signaling in individual cells. It cannot be excluded that some other GPCR (or transporters or channels) with 2 glycan chains at the appropriate distance (about 10% of GPCRs fall in this category) might contribute to a minor proportion of meningococcus signaling.

To further support the predominant role of the β 2AR we investigated ezrin recruitment in β 2AR KO hCMEC/D3 cells derived from parental hCMEC/D3 cells using CRISPR/Cas9 approach (Supplementary fig 4). A 15% of residual ezrin recruitment is observed in these cells upon *N. meningitidis* infection, supporting the existence of some β 2AR independent or residual signaling. Noteworthy, this value is comparable with what is observed in the HEK-293 reconstituted model (see Figure 1d). This point has been addressed in the discussion section of the revised manuscript.

Annex 1

Effect of overexpressed CD147 in meningococcus-promoted β 2AR signalling in reconstituted HEK-293 cells.

Annex 2

hCMEC/D3 transfected with β 2AR-YFP, incubated with 10 μ M isoproterenol after meningococcus infection

HEK-293 reconstituted cells (as in Fig. 1) treated or not with 10 μ M isoproterenol for 30 min after infection

Reviewers' Comments:

Reviewer #1:

Remarks to the Author:

The authors have provided excellent characterization of the sialoglycans on beta 2-arrestin and have addressed my critiques thoroughly. Very minor editorial comments:

1. I do not see any reference to Supplemental Fig 8e in the text – I think some description of this experiment is needed.
2. Line 609 – should be PNGase (not PNGAse)
3. Spelling of sialylation in Supplemental Figure 7a – sialylation (not sialilation)

Reviewer #2:

Remarks to the Author:

The Authors satisfactorily addressed the question raised.

Reviewer #3:

Remarks to the Author:

In their revised manuscript Virion et al addressed comments of the other referees, but did relatively little to address concerns about claims regarding signaling through the β 2AR. They state that “classical pharmacological approaches [...] are not appropriate here”. This is puzzling, since conventional β -arrestin recruitment assays and cAMP accumulation assays are straightforward and simple experiments requiring little specialized equipment. These experiments can be performed with commercially available kits (PRESTO-TANGO, DiscoverX kits, etc.), or they can be performed on a fee-for-service basis through CROs. In my view, all claims regarding activation of the β 2 receptor and biased signaling are unsupported in the absence of a standard widely accepted β -arrestin recruitment assay in dose-response format with appropriate controls. In the absence of this data I would not recommend this manuscript for publication.

The authors do include a new figure panel (3E) that reports β -arrestin2 recruitment at a single time point. This is difficult to evaluate because it does not appear to be described in the methods section, and it is unclear how it was quantified. Fusing GFP to the C-terminus of β -arrestin may affect its function since the C-terminal tail is a key regulator of arrestin-mediated internalization (see Schmid et al 2006 Plos Biol. 4: E262 for example). The exact design and validation of this construct would be required to evaluate the suitability of this assay. In addition, standard controls are currently omitted and should be included. These include showing an antagonist blockade negative control to ensure specificity of the observed interaction, and a β 2AR agonist (isoprenaline or BI167107 for example) as a positive control.

Lastly, the authors point out that saturating propranolol does not inhibit meningococcal signaling in their previous paper. Beta2 receptor cannot be activated when saturating propranolol is present (it is an inverse agonist and sterically clashes with activated receptor conformations), which seems to imply that the purported signaling does not go through β 2 receptor after all. This of course compromises many of the claims of the prior Cell paper as well as the current work. Since the authors' Cell paper was published nearly ten years ago, has any other group observed signaling or even interaction between meningococcus and β 2 receptor? If not, would this be cause for concern that much of what is under investigation here is artifactual?

Minor point: In supplementary figure 4 it would be helpful to include chromatograms from Sanger sequencing instead of just base calls. Confirming the lack of beta2 receptor by radioligand binding would be a straightforward and simple experiment (less than 4 hours of work for an experienced lab) and could also be done through a CRO for a modest fee. This would be an ideal control to confirm absence of functional beta2 receptor protein.

Reviewer #4:
None

We thank the Reviewers for detailed examination of our manuscript and constructive comments. We feel that the manuscript is much stronger as a result of their input.

Reviewer #1 (Remarks to the Author):

The authors have provided excellent characterization of the sialoglycans on beta 2-AR and have addressed my critiques thoroughly. Very minor editorial comments:

1. I do not see any reference to Supplemental Fig 8e in the text – I think some description of this experiment is needed.

Experiment described top of page 10

2. Line 609 – should be PNGase (not PNGase)

corrected

3. Spelling of sialylation in Supplemental Figure 7a – sialylation (not sialilation)

corrected

Reviewer #2 (Remarks to the Author):

The Authors satisfactorily addressed the question raised.

Reviewer #4

Data provided in the Annex should be added to the supplementary data as it shows that meningococcus infection reduces sympathetic beta-adrenergic receptor responses.

Data added in the Supplementary Figure 10 and discussed in the discussion section

Reviewer #3

In their revised manuscript Virion et al addressed comments of the other referees, but did relatively little to address concerns about claims regarding signaling through the β 2AR. They state that “classical pharmacological approaches [...] are not appropriate here”. This is puzzling, since conventional β -arrestin recruitment assays and cAMP accumulation assays are straightforward and simple experiments requiring little specialized equipment. These experiments can be performed with commercially available kits (PRESTO-TANGO, DiscoverX kits, etc.), or they can be performed on a fee-for-service basis through CROs. In my view, all claims regarding activation of the β 2 receptor and biased signaling are unsupported in the absence of a standard widely accepted β -arrestin recruitment assay in dose-response format with appropriate controls. In the absence of this data I would not recommend this manuscript for publication.

The concerns raised by the reviewer would be valid from the perspective of a classical ligand GPCR interaction. As demonstrated by our multiple previous publications in the field, the approaches indicated by the reviewer are well mastered and routinely performed in our laboratory. Apologizing for the length of our demonstration, we would like to explain why the classical paradigm of ligand-GPCR interaction is not appropriate here and, consequently why we have used alternative methods.

As mentioned by the reviewer typical dose response assays to elicit a functional response of the β 2AR are conducted with 100 nM-10 μ M isoproterenol. These concentrations correspond to 6×10^{13} to 6×10^{15} agonist molecules in solution per ml. In our current bacterium – host cell interaction paradigm, the same number of cells (about 100-150 10^3) per assay are incubated with a maximal multiplicity of infection (MOI) of 100 (i.e. $1-1,5 \times 10^7$ bacteria), higher MOI flooding cells with PAMPs (Pathogen-associated molecular patterns, becoming toxic for the cells in culture). Among them only a small proportion will adhere and form colonies at the surface of endothelial cells. Once bacteria have adhered via CD147, they interact with and activate receptors via bundles of polymeric pilins within the bacterial pili (5-10 per bacterium). Only receptors in the immediate proximity of these structures can be bound and activated by the pathogen whereas soluble ligands can interact with all available surface receptors (according to the K_D).

Therefore, the equilibrium conditions and concentrations governing the interaction with the receptor are extremely different between a soluble agonist and bacterial organelles.

Moreover, the agonist interact with the orthosteric binding site inducing a conformational change of the receptors whereas we demonstrated that pili exert a traction force on the receptor applied via the N-glycans. Because of this very particular mode of activation (somehow reminiscent of what has been reported for adhesion GPCRs) free purified pilins, although they bind to the β 2AR (see our HTRF data), cannot activate the receptor because they do not produce the mechanical forces required to lead their activation.

In the previous versions of the manuscript we wrote in some sentences that receptor was activated by “pilins” since PilV and PilE are the binding ligands that interact with receptor N-terminus. Although formally correct, we realize that this might be confusing since, as explained above, receptor activation is only possible via functional pili (not via pilins in solution). We reworded the corresponding sentences by substituting pilins by pili.

A second constraint in the experimental design is related to the pathogenic nature of the bacterium. This issue implies that all the experiments are performed within a protected area containing the culture facility, the confocal microscope and the basic lab equipment. Samples for biochemical analysis (PAGE and blot, cAMP assays...) are first treated with appropriate detergents or compound inactivating pathogenic bacteria. Because of this technical constrains, people working on meningococcus interaction with host cells has developed for more than 30 years specific assays, based on IF-based visualization of the accumulation of signaling molecules, such as ezrin, actin, p120 catenin, VE-cadherin (beta arrestin and β 2AR, in our case) under bacterial colonies. In addition, in the field of host pathogen interaction, IF experiments, showing the pathogen-induced recruitment of signaling molecules in host cells, are the gold standard. See for example: Roy NH, et al *J Virol* 88:7645-58, 2014 for ezrin and pERM/HIV; Valencia-Gallardo C et al, *Cell Rep* 26:921-932.e6, 2019 for talin/shigella; Sun CH et al, *EMBO J.* 2017 Sep 1; 36: 2567–2580, 2017 for InsP3 receptors/shigella; Zhang Y et al, *Nat Immunol.* 20:433-446, 2019 for Nod-like receptor X1/listeria; Saadat I et al, *Nature* 447:330-3, 2007 for PAR1-MARK kinase/H. pylori; Rajabian T et al, *Nat Cell Biol* 11:1212-8, 2009 for ezrin, actin/listeria.

The cAMP assay requested by the reviewer has actually been published in our Cell paper (See Annex 1 at the end of this letter) in the experimental context explained above. We used a sensitive cAMP reporter assay, which showed a nice cAMP-dependent signal upon activation with saturating isoproterenol and no significant signal with bacteria at the highest possible MOI of 100:1. However, because under the same conditions we could observe a translocation of β -arrestin under the colonies we concluded (based on this finding and many other controls, see below) that the pathogen induces a β -arrestin biased activation of the β 2AR.

We understand that some authors prefer not to use the term biased signaling when bias cannot be calculated using the operational model (which is the case in our study since no concentration

response curves can be obtained for the cAMP production). We therefore propose to remove the term “biased signaling” to replace it by “functional selectivity”.

In addition to our explanations indicating why more traditional approaches cannot be used here, we would like to summarize the body of experimental evidence supporting the conclusion that the β 2AR is activated by meningococcus:

1. The induction of signaling events in human cells depends on the presence of β 2AR at the cell surface. Inhibition of β 2AR expression with siRNA or after knock out (by CRISPR/Cas9) reduces the signaling of meningococcus *in vitro* to residual levels. Incubation with isoproterenol to induce endocytosis before (Coureuil M et al, *Cell* 143:1149-1160, 2010) or during infection (see Supplementary Fig 10 in the revised version of the manuscript) largely inhibits meningococcus-induced signaling. In the present manuscript several point mutations of the β 2AR targeting the Asn residues or the receptor N-terminus (suppression of either glycosylation site or both or changing the space between the glycosylation sites) inhibit meningococcus signaling. We successfully set up a gain of function experimental approach in human cells (Coureuil M et al, *Cell* 143:1149-1160, 2010). Under basal conditions, HEK-293 cells cannot be infected by meningococcus. However, upon reconstitution of these cells with β 2AR (and NOT with other control GPCRs), β -arrestins (and NOT without β -arrestins) and CEACAM (to provide a surrogate adhesion receptor to the pathogen), not only is signaling reconstituted but also several phenotypic changes typically observed in endothelial cells infected by meningococcus are produced, such as the actin-rich protrusions which surround colonies and stabilize them under flow.
2. We reported that the “functional” receptor for meningococcus in endothelial cells is a complex of 3 proteins: CD147, the early adhesion receptor, in complex with the β 2AR, stabilized by α -actinin-4. In the article by Maissa et al (*Nat. Commun.* 8:15764, 2017) we have demonstrated that this particular organization (i.e. the presence of the 3 components of the complex) is absolutely necessary for a proper host-pathogen interaction.
3. In the present manuscript we report the specific interaction of bacterial pilins with the receptor N-terminus by an HTRF approach. If soluble pilins cannot on their own activate the receptor (for the mechanical reasons explained in our current manuscript), once attached to an unrelated inactive pathogen (staphylococcus) they can activate β 2AR signaling and β -arrestin translocation (Coureuil M et al, *Cell* 143:1149-1160, 2010)
4. In the present article we elucidated the mechanism by which the β 2AR can be mechanically activated by the traction forces exerted by bacterial pili.
5. Finally, we show in the Annex 3 below the time course of β -arr recruitment to the β 2AR following meningococcal infection, using Bioluminescence Resonance Energy Transfer

The authors do include a new figure panel (3E) that reports β -arrestin2 recruitment at a single time point. This is difficult to evaluate because it does not appear to be described in the methods section, and it is unclear how it was quantified. Fusing GFP to the C-terminus of β -arrestin may affect its function since the C-terminal tail is a key regulator of arrestin-mediated internalization (see Schmid et al 2006 Plos Biol. 4: E262 for example). The exact design and validation of this construct would be required to evaluate the suitability of this assay. In addition, standard controls are currently omitted and should be included. These include showing an antagonist blockade negative control to ensure specificity of the observed interaction, and a β 2AR agonist (isoprenaline or BI167107 for example) as a positive control.

A more detailed description of the methods for quantifying the very particular mode of activation of the β 2AR, based on counting under the microscope the recruitment of signaling molecules under the colonies, has been included in the methods section. The principal method used in most figures is based on the visual counting the % of colonies recruiting the proteins of interest. The second (used for example in Fig 6b) quantifies the total fluorescence signal associated to the protein of interest

below bacterial colonies. As shown in Fig 6a and 6b for ezrin recruitment, the 2 methods globally provide the same information with two different experimental setups. In the Annex 2 below, we have used the second approach to examine β -arrestin translocation: the average fluorescence reflecting β -arrestin recruitment under the colonies was measured pixel by pixel and plotted as a function of the area of the colony. This area is variable. Indeed, an important point to consider, is that once inoculated at 37°C bacterial colonies are not forming evenly. Some bacteria remain single and do not divide in one hour, whereas growing colonies of different size coexist. Globally, the IF signal associated with β -arrestin translocation below the colony grows as a function of the area of the colony, since the amount of β 2AR in proximity of bacterial pili increases with the size of the colony, assuming that the β 2AR is evenly distributed at the cell surface before infection. In case of pre-incubation with isoproterenol to induce β 2AR endocytosis, the IF signal corresponding to β -arrestin-YFP accumulation under the colonies is globally reduced whatever is the size of the colony. In these experiments the isoproterenol-induced endocytosis of the β 2ARs that are not in direct contact with the colony represents a specific positive activation control of the β 2AR (as visible in the Supplementary Figure 10). Finally, to address the issue of the “single time point” we show in the Annex 3 why our method is pertinent using a demonstrative bioluminescence resonance energy transfer (BRET) approach in the HEK-293 cell reconstituted system. The BRET approach has extensively been used in our laboratory to measure proximity between proteins of interest in live cells and BRET has specifically been used in the literature to monitor β -arrestin translocation to activated GPCRs (*J Biol Chem.* 2003;278:41541-51 (2003); *Proc Natl Acad Sci U S A.* 105:13656-61 (2008); *J Biol Chem.* 293:6161-6171 (2018)). In the panel (a) of Annex 3 the 5 min isoproterenol-promoted translocation of β arr2-Rluc (expressed at constant concentration) to the β 2AR-YFP led to a saturable BRET signal for increasing levels of the receptor. In the absence of agonist only a bystander linear BRET was visible. The asterisk indicates the experimental conditions used in subsequent experiments. In panel (b) BRET were measured in cells expressing β arr2-Rluc alone or co-expressed with the β 2AR-YFP (red histograms) or the AT1R-YFP (blue histograms). A BRET signal was promoted by incubation with the respective agonists. Note that the BRET signal obtained with equivalent (based on the measurement of respective fluorescence signals) concentrations of AT1R-YFP was much higher than for β 2AR-YFP. A different distance from the BRET donor and acceptor in the complex and the fact that β arr interaction with AT1R is more stable than with β 2AR likely explain this finding. In the case of β 2AR-YFP, 1h pre-incubation with Isoproterenol abolished the BRET (internalized and recycling β 2AR is known to dissociate from β arrs). Panel (c) shows time course translocation of β arr2-Rluc to the indicated receptors upon addition of meningococci. BRET values are represented as the % of the maximal BRET obtained at 5 min with the respective agonists. No significant BRET was induced by bacteria in cells expressing the AT1R-YFP, despite the higher BRET signal elicited by Ang II. In cells expressing the β 2AR-YFP the BRET increased slowly with time reaching a stable plateau at 40 min. Thus the 60 min time used throughout the article corresponds to the maximal stable recruitment of β -arrestins under the colonies. Note that the BRETmax obtained with bacteria is about 25% of that induced by the agonist, since bacteria induce the translocation of β arr2-Rluc only in the cells directly in contact with the colonies.

It is true that the C-terminal tail of β -arrestin is a key regulator of β -arrestin. Nevertheless, fusing GFP (or other tags) to the C-terminus of β -arrestin has been widely used in assays (such as the BRET assays above) investigating GPCR activation and/or β -arrestin functions, since the seminal publication of Marc Caron group, which described this approach: “Barak LS et al. A beta-arrestin green fluorescent protein biosensor for detecting G protein-coupled receptor activation, *J Biol Chem*, 272: 27497-27500, 1997” (~350 citations according to WOS). This assay is particularly well adapted in our case since, as indicated by these authors, “It provides a real-time and single cell based assay to monitor GPCR activation and GPCR-G protein-coupled receptor kinase or GPCR-arrestin interactions”. Moreover, the article cited by the reviewer (*Schmid et al 2006 Plos Biol.*) was co-signed by one of our collaborators (A. Benmerah) who precisely used a β -arrestin-GFP construct provided by us to

perform some of the experiments included in that article. In addition, most authors of that article were our co-authors in a subsequent article in which our β -arrestin GFP constructs were used to investigate specifically the interaction of β -arrestin with the adaptor protein AP2 (Burtsey et al. *Traffic* 2007; 8:914–931).

Finally, the Presto-tango or Discoverex assays recommended by the reviewer are similarly based on β -arrestin fusions at its C-terminus (Tev protease and β -galactosidase, respectively). Our BRET-recruitment assay in the Annex 3 is based on the same general principle.

Lastly, the authors point out that saturating propranolol does not inhibit meningococcal signaling in their previous paper. Beta2 receptor cannot be activated when saturating propranolol is present (it is an inverse agonist and sterically clashes with activated receptor conformations), which seems to imply that the purported signaling does not go through β 2 receptor after all. This of course compromises many of the claims of the prior Cell paper as well as the current work.

It is true that in one figure of the Cell paper we showed that 100nM propranolol could not block ezrin recruitment by meningococcus. We interpreted this finding as an indication of some allosteric activation of the β 2AR (largely confirmed in our current manuscript). We disagree with the reviewer when he/she concludes that this finding implies that the “the purported signaling does not go through β 2 receptor after all”, since this statement is disproved by published data. Several inverse agonists were reported to inhibit the receptor conformation capable of coupling to the G protein and, at the same time, promote another conformation that activates another pathway. For example carvedilol is able to stabilize a receptor conformation, which, although uncoupled from G_s , is nonetheless able to stimulate β -arrestin-mediated signaling (Wisler, JW et al, *PNAS* 104:16657-62, 2007). Propranolol itself was reported to simultaneously act as an inverse agonist of the β 2AR through a G_s -coupled mechanism, while stimulating the p42/44-MAP kinase pathway through an alternative G-protein-independent mechanism (Baker et al *Mol Pharmacol.* 64:1357-69, 2003) and promoting β -arrestin translocation to the receptor (Azzi et al, *PNAS* 2003 100:11406-11, 2003). These findings demonstrate that other active conformations of the β 2AR are possible (such as those promoted by meningococcus) while propranolol occupies the orthosteric binding site.

Since the authors’ Cell paper was published nearly ten years ago, has any other group observed signaling or even interaction between meningococcus and β 2 receptor? If not, would this be cause for concern that much of what is under investigation here is artifactual?

For many studies exploring new concepts with a limited number of active laboratories in the specific field (a given pathogen interaction with host cell receptors) and/or very specific experimental approaches it can take time before the appropriation of the topic by a large audience. Many seminal structural studies on specific GPCRs have never been reproduced by others after several years. Our articles have been read, as testified by the number of citations. If people would have not been able to reproduce the data, it certainly would have been claimed somewhere. In addition, the reviewer apparently missed an article published by our major competitor, Guillaume Duménil of the Institut Pasteur who used isoproterenol to down modulate β 2AR prior meningococcus infection and concluded “...This suggests that the β 2 adrenergic signaling pathway is necessary to attain the full extent of actin reorganization rather than being involved in the initiation of actin reorganization”. (Soyer et al. *Cellular Microbiology* (2014) 16, 878–895).

Finally, not only are our articles on this topic well cited, our work has been presented in prestigious conferences attended by specialists of both microbiology and GPCR fields (Gordon and Keystone conferences, GPCR retreats, ASPET symposium..) and the reproducibility of our data has never been an issue.

If any of our data could have not been reproduced by this reviewer or one of his (her) collaborators, we are ready to do our best to address the issue as soon as possible, share reagents and methods and even host people in our laboratory to reproduce the experiments.

Minor point: In supplementary figure 4 it would be helpful to include chromatograms from Sanger sequencing instead of just base calls. Confirming the lack of beta2 receptor by radioligand binding would be a straightforward and simple experiment (less than 4 hours of work for an experienced lab) and could also be done through a CRO for a modest fee. This would be an ideal control to confirm absence of functional beta2 receptor protein.

The requested chromatograms have been included in the figure. We agree that radio-ligand binding assays are straightforward and we have largely used them for decades. Unfortunately, nowadays the use of radioactive compounds is very restricted and controlled in France, and we should justify the fact that there are not other methods to address a specific question (which is clearly not the case here).

Annex 1:

From Coureuil M et al, Cell 143:1149-1160, 2010

Supplemental Figure S2D: **Production of cAMP in hCMEC/D3 cells assessed by a cAMP reporter assay in response to bacterial adhesion or β -adrenergic ligands.** The assay is described in Supplemental Experimental Procedures. *WTNm*, wild-type 2C43 bacteria; *WTNm propra*, wild-type 2C43 bacteria in the presence of 100nM propranolol; *Pile-Nm*, 2C43 Δ PilE bacteria devoid of type IV pili. *iso*, 10 μ M isoproterenol; *iso - propra*, 10 μ M isoproterenol and 100nM propranolol; ***($p < 0.001$).

Supplemental Experimental Procedures

CRE-Luciferase assay. Luciferase activity in cell lysates was measured using the Luciferase Reporter Assay System (Promega) according to manufacturer instructions. Briefly, hCMEC/D3 cells were transfected with the 4xCRE-Luciferase plasmid (Stratagene) the day before the experiment. Cell infection was performed as described above at a MOI of 100. After 2h of infection or incubation with drugs or ligands, cells were washed twice with ice-cold PBS, and lysed using the Luciferase Cell Culture Lysis Reagent. Lysates were centrifuged at 12,000xg for 2min at 4°C. Supernatants were tested for luciferase expression using a VICTOR3 multilabel counter (Perkin-Elmer).

Annex 2:

	βarrestin2-GFP recruitment (median of AU)	βarrestin2-GFP recruitment (mean of AU ± SEM)	Mann-Whitney U test
control	305.7	647.6 ± 143.5	
+iso 10μM	14.67	123.6 ± 31.9	p=0.0061

Endothelial hCMEC/D3 cells were transfected with the βarrestin2-GFP construct, using AMAXA™ nucleofection, and infected the next day with wild type *N. meningitidis* with or without isoproterenol pre-incubation (iso 10μM). Cells were then processed for the analysis of βarrestin2-GFP recruitment below bacterial colonies as in Fig. 6b, c. The intensity of βarrestin2-GFP recruitment was quantified for each colony individually (see Supplementary Methods) and expressed as arbitrary intensity below colonies. Each dot represents one colony. Linear regression was calculated for both conditions using GraphPad Prism 8.

Annex 3:

Time course of β arr2 translocation to meningococcal colonies in HEK-293 cells

a) HEK cells (1×10^5 per well of a 12-well plates) were transfected with 20 ng plasmid DNA coding for β arr2-Rluc (the BRET donor) and increasing amounts of the BRET acceptor plasmid (β 2AR-YFP, 0-600 ng per well). Twenty-four hours after transfection, cells were resuspended in FluroBrite™ DMEM (Gibco) supplemented with 10% SVF and L-glutamine and distributed in 96-well plates (PerkinElmer plates; 10^4 cells per well). After addition of the luciferase substrate, coelenterazine h (5 μ M final

concentration), luminescence at 485 nm and fluorescence at 530 nm were measured simultaneously in a Mithras2 LB 943 plate reader. The BRET ratio was calculated as detailed in Achour L. et al. **Methods Mol Biol.**, 756:183-200 (2011): $([\text{emission at 530 nm}/\text{emission at 485 nm}] - [\text{background at 530 nm}/\text{background at 485 nm}])$, where background corresponds to signals in cells expressing the Rluc fusion protein alone under the same experimental conditions. For better readability, results were expressed in milli-BRET units (mBRET), 1 mBRET corresponding to the BRET ratio multiplied by 1000. BRET ratios were plotted as a function of $(\text{YFP})/(\text{Rluc})$, where YFP is the fluorescence signal at 530 nm after excitation at 485 nm (subtracted of the same value in cells expressing the β arr2-Rluc alone. β_2 AR stimulation assays was achieved by the addition of isoproterenol at a final concentration of 10 μM and the energy transfer was measured after 5 min. **b**) Same experiment as in (a) but at a single BRET acceptor concentration (conditions indicated in panel a by the asterisk). For the AT1R-YFP controls β arr2-Rluc was transfected at 20 ng and AT1R-YFP at 300 ng. BRET were measured in cells expressing β arr2-Rluc alone or co-expressed with the β_2 AR-YFP (red histograms) or the AT1R-YFP (blue histograms). Agonist-promoted BRET signal was measured upon incubation with the respective agonists. **c**) For measuring meningococcus-induced translocation of β arr2, the cells were also co-transfected with human CEACAM-1 (400 ng), β arr2-Rluc and the indicated BRET acceptors. 24 h after transfection the cells were seeded in 96-well plates 10^4 cells per well. Bacterial colonies were harvested from GCB plates and were adjusted to $\text{OD}_{600} = 0.05$ and grown for 2 hours at 37°C in 5% CO_2 incubator under constant shaking in in FluroBrite™ DMEM (Gibco) supplemented with 10% SVF and L-glutamine. After two hours the OD was measured ($\text{OD}_{600} = 0.2$) the bacteria were inoculated at a MOI of 100:1. Coelenterazine h was added at the indicated times and BRET signals measured immediately after. The indicated net BRET values were obtained after subtraction of BRET values obtained with non-signaling Pili-deficient bacteria. The maximal BRET values were measured on the same cells after 5 min incubation with respective receptor agonists as in **(b)** and *Nm*-induced BRET was expressed as the % of the maximal BRET value. Experiments were performed in triplicate.

Reviewers' Comments:

Reviewer #3:

Remarks to the Author:

In their revised manuscript, Virion et al continue to claim that meningococcus activates beta2 receptor, while not providing sufficient data to support this claim, in my view. In the response to reviewers the authors provide extensive explanations for not including new data to support their claims, but in the absence of standard signaling assays and pharmacological controls I am not convinced that meningococcus actually activates beta2 receptor. As noted, these assays can be performed by CROs on a fee for service basis, which should eliminate entirely concerns about radioactive binding assay restrictions in France, etc.

Answers to Rev#4 comments

In the current manuscript, the meningococcus bacteria mediates ezrin recruitment to the beta-2 adrenergic receptor through β -arrestin-dependent mechanisms that seems to be independent of the classical G-protein and cAMP axis. Thus, the requirement to assess cAMP may not be required in this given landscape.

Though the determination on the requirement of cAMP analysis is much simpler, it is more complex with regards to the concern of β -arrestin raised by the reviewer 3. I agree with the comprehensive response of the authors in the context of β -arrestin recruitment assay given the non-canonical activation of β -adrenergic receptor. Also, I believe that these set of studies are appropriate to show that β -arrestin is involved in the nucleation with Ezrin and β -adrenergic receptor that occurs through non-canonical mechanisms through mechano-transduction mediated by pili. In this context, there is ample evidence of studies (from other labs) showing stretch-mediated activation of receptor can occur through β -arrestin dependent pathways and to a great extent aligns with the current studies showing that meningococcal pili could mediate activation through β -adrenergic receptor- β -arrestin axis.

Furthermore, data showing that pre-treatment with ISO (isoproterenol) significantly reduces ezrin recruitment is another key piece of evidence supporting the idea that meningococcal bacteria engages β -adrenergic receptors for mediating its effects. Given that ISO is a full agonist for β -adrenergic receptors thus, pre-engaging the β -2 adrenergic receptors now impairs the effects of the meningococcal bacteria.

It would however be important for authors to rephrase their response with regards to the beta-arrestin especially the statement

“However, upon reconstitution of these cells with β 2AR (and NOT with other control GPCRs), β -arrestins (and NOT without β -arrestins) and CEACAM (to provide a surrogate adhesion receptor to the pathogen)”.

In this context, the authors state that they have used cell systems with no β -arrestins but at best, I could only see overexpression in HEK 293 cells with β -arrestin wherein, exogenous transfection of β -arrestin leads to elevated ezrin recruitment in response to centrifugal forces (Fig. 7). However, in Fig. 7 the β -arrestin (β -arrs) in the left of panel only represents exogenous transfection/addition and not the knockdown or absence of endogenous β -arrestin (but this what is the annotated Fig. 7 depicts) which needs to be clarified given that HEK 293 cells do have appreciable levels of endogenous β -arrestin. In recognition, it would be important for the authors to temper down the “NOT without β -arrestin” statement. Also, it is important for the authors to discuss why the β -arrestin and ezrin recruitment is not observed at baseline (ie without transfection) in HEK 293 cells following centrifugal forces. Is there a threshold requirement that mediates ezrin recruitment response given endogenous expression of β -arrestin? How would knockdown or knockout of β -arrestin alter these observations? This needs to be thought through by the authors in their overall discussion.

We would like to thanks Rev#4 for the additional reviewing and for the appreciation of our work. The reviewer is right: although the critical role of endogenous β -arrestins was demonstrated in early siRNA experiments in immortalized endothelial cells (Coureuil et al, Cell 2010), there is clearly a threshold of β -arrestin concentration, which has to be reached to observe ezrin recruitment below colonies and below WGA-coated beads. This issue is discussed in the final version of the manuscript.

However, I believe that the authors have provided evidence of β -arrestin involvement in these studies showing that it engages β -adrenergic receptor and β -arrestin axis in a non-canonical manner. Although knock down of β -arrestin would provide unequivocal evidence but given the absence of ezrin recruitment in HEK 293 cells with endogenous β -arrestin recruitment, it may not provide insights on the non-canonical role of β -arrestin in this

process. Given this conundrum, it would be important for the authors to rationally assess their observations and test whether there is a threshold requirement for β -arrestin in ezrin recruitment/engagement as only a small fraction of the β -adrenergic receptors could be engaged by the pili.

In summary, given the identification of the non-canonical mechanisms underlying the engagement of β -adrenergic receptors, the evidence currently provided outweighs experimental concerns regarding the need for classical G-protein signaling mechanisms mediated by meningococcal bacteria in my opinion.